# LiNeS: Post-training Layer Scaling Prevents Forgetting and Enhances Model Merging

**Ke Wang**[*]
EPFL
k.wang@epfl.ch

**Nikolaos Dimitriadis**[*]
EPFL
nikolaos.dimitriadis@epfl.ch

**Alessandro Favero**
EPFL
alessandro.favero@epfl.ch

**Guillermo Ortiz-Jimenez**
Google DeepMind
gortizj@google.com

**François Fleuret**
University of Geneva, Meta FAIR
francois.fleuret@unige.ch

**Pascal Frossard**
EPFL
pascal.frossard@epfl.ch

## Abstract

Fine-tuning pre-trained models has become the standard approach to endow them with specialized knowledge, but it poses fundamental challenges. In particular, *(i)* fine-tuning often leads to catastrophic forgetting, where improvements on a target domain degrade generalization on other tasks, and *(ii)* merging fine-tuned checkpoints from disparate tasks can lead to significant performance loss. To address these challenges, we introduce LiNeS, Layer-increasing Network Scaling, a post-training editing technique designed to preserve pre-trained generalization while enhancing fine-tuned task performance. LiNeS scales parameter updates linearly based on their layer depth within the network, maintaining shallow layers close to their pre-trained values to preserve general features while allowing deeper layers to retain task-specific representations. In multi-task model merging scenarios, layer-wise scaling of merged parameters reduces negative task interference. LiNeS demonstrates significant improvements in both single-task and multi-task settings across various benchmarks in vision and natural language processing. It mitigates forgetting, enhances out-of-distribution generalization, integrates seamlessly with existing multi-task model merging baselines improving their performance across benchmarks and model sizes, and can boost generalization when merging LLM policies aligned with different rewards via RLHF. Our method is simple to implement, computationally efficient and complementary to many existing techniques. Our source code is available at github.com/wang-kee/LiNeS.

## 1 Introduction

Pre-trained models have become the backbone of modern machine learning pipelines (Bommasani et al., 2021; Touvron et al., 2023). Their introduction has shifted the paradigm from end-to-end training to fine-tuning (Zhuang et al., 2020), leading to the proliferation of thousands of fine-tuned checkpoints derived from a few foundation models (Rombach et al., 2022; Team et al., 2023). To improve downstream performance across multiple tasks or align with multiple preferences (Singh & Jaggi, 2020; Matena & Raffel, 2022; Ilharco et al., 2023; Yadav et al., 2023; Ramé et al., 2024a), model merging techniques combine available checkpoints, avoiding the costly process of joint fine-tuning (Ilharco et al., 2023; Yadav et al., 2023). However, specializing models introduces trade-offs, such as the forgetting of previously acquired knowledge (Aghajanyan et al., 2021) – a phenomenon known as *catastrophic forgetting* (McCloskey & Cohen, 1989). Furthermore, merging checkpoints fine-tuned on different tasks can lead to significant performance degradation due to task interference (Yadav et al., 2023; Wang et al., 2024).

---

[*]Equal Contribution

To mitigate catastrophic forgetting, many works propose regularizing the fine-tuning process (Aghajanyan et al., 2021; Kumar et al., 2022; Gouk et al., 2021; Razdaibiedina et al., 2023). Leveraging the insight that shallow layers capture generalizable representations (Yosinski et al., 2014; Neyshabur et al., 2020), Howard & Ruder (2018); Dong et al. (2022) apply lower learning rates to the shallow layers to retain general features. However, modifying the fine-tuning process can be complex and computationally expensive. This motivates the development of post-training model editing and model merging methods that directly edit the checkpoints in the weight space. For instance, Wortsman et al. (2022b); Ramé et al. (2022) mitigate catastrophic forgetting by interpolating weights between pre-trained and fine-tuned models. In multi-task settings, Yadav et al. (2023); Wang et al. (2024) propose methods to reduce interference among tasks when merging multiple checkpoints. Yet, significant performance degradation persists when merging multiple models, leaving this as an open challenge.

Most model merging methods, however, treat all layers equally, overlooking the earlier insight that shallow layers should remain close to their pre-trained weights to avoid losing the general representations they encode. In this paper, we explore whether this insight can be leveraged post-training. We find that reducing the magnitude of shallow-layer updates after fine-tuning can retain single-task performance gains while significantly mitigating forgetting.

We propose LiNeS, **L**ayer-**i**ncreasing **Ne**twork **S**caling, a post-training, plug-and-play method that directly edits the residual, i.e., the difference between the fine-tuned and pre-trained checkpoint, by applying a scaling coefficient that linearly increases with layer depth. This scaling effectively preserves the general features captured in the shallow layers of the pre-trained model while retaining task-specific features in the deep layers of the fine-tuned model. Moreover, we extend LiNeS to the multi-task model merging setting, where contributions from one task distort the general features also required by other tasks. By preserving the general features in the shallow layers, LiNeS mitigates task interference and improves multi-task performance.

LiNeS demonstrates remarkable performance on diverse test scenarios and is orthogonal to existing post-training merging algorithms. It modifies the fine-tuned checkpoint to consistently retrieve nearly full performance on the fine-tuned task while significantly restoring generalization on other tasks. Furthermore, it can be seamlessly integrated with existing weight interpolation methods for improving out-of-distribution generalization (Wortsman et al., 2022b). When merging multiple models, LiNeS improves baseline methods for merging checkpoints fine-tuned on multiple tasks in both computer vision and NLP benchmarks (Ilharco et al., 2023; Yadav et al., 2023; Wang et al., 2024) and also enhances performance when merging checkpoints fine-tuned on the same task (Wortsman et al., 2022a) and merging LLM policies aligned with different rewards (Ramé et al., 2024a) via Reinforcement Learning with Human Feedback (RLHF) (Christiano et al., 2017).

Our contributions are as follows:

- We propose LiNeS, a post-training editing technique that preserves the zero-shot generalization of pre-trained models while retaining fine-tuned knowledge by applying layer-wise scaling on parameter updates. For example, in image-classification tasks with CLIP ViT-B/32 checkpoints, LiNeS maintains on average 99.8% of performance on the fine-tuned task while preserving 97.9% performance of the pre-trained model on other control tasks, effectively mitigating catastrophic forgetting.

- We demonstrate that LiNeS significantly enhances multi-task model merging baselines, consistently improving performance across benchmarks and architectures in both vision and NLP domains. For instance, we observe a 3.1% and 4.0% improvement over Task Arithmetic (Ilharco et al., 2023) and Ties-merging (Yadav et al., 2023) respectively, for a 20-task computer vision benchmark with ViT-L/14.

- We show that LiNeS can be applied to enhance existing weight interpolation methods across various scenarios, improving out-of-distribution generalization, merging multiple checkpoints fine-tuned on the same task with different hyper-parameter configurations, and merging LLM policies aligned with different rewards.

Our proposed method is simple to implement[1], orthogonal to many existing approaches, and improves performance in a wide variety of settings.

---

[1]PyTorch pseudo-code in Appendix A.

## 2 RELATED WORK

**Representation collapse and regularized fine-tuning**    Pre-trained models such as CLIP exhibit strong zero-shot performance across diverse data distributions due to the robust and transferable feature representations learned during pre-training (Radford et al., 2021; Jia et al., 2021). However, fine-tuning on specific tasks often harms the zero-shot generalization performance on distributions different from the fine-tuning domain (Wortsman et al., 2022b; Goyal et al., 2023; Aghajanyan et al., 2021). This degradation arises from the distortion of pre-trained features during fine-tuning (Kumar et al., 2022), a phenomenon referred to as *representation collapse* by Aghajanyan et al. (2021). To mitigate representation collapse, many works have proposed to regularize the fine-tuning process to preserve the general pre-trained features (Kumar et al., 2022; Goyal et al., 2023; Gouk et al., 2021; Zhang et al., 2022; Razdaibiedina et al., 2023; Shen et al., 2021; Lee et al., 2022). Some of these approaches take into account that different layers of a model learn distinct features, with the shallower layers capturing more general features and deeper layers specializing in task-specific representations (Neyshabur et al., 2020; Yosinski et al., 2014; Adilova et al., 2024). Specifically, they apply layer-wise learning rate decay, preserving more of the pre-trained features in the shallow layers while allowing deeper layers to specialize for the target domain (Clark et al., 2020; Bao et al., 2022; Dong et al., 2022; Howard & Ruder, 2018; Zhang et al., 2021). However, modifying the fine-tuning process is orders of magnitude more computationally expensive compared to post-training merging methods.

**Weight interpolation and model merging**    Garipov et al. (2018); Draxler et al. (2018) showed that two solutions derived from separate training runs can be connected by nonlinear paths of low loss, while *linear mode connectivity* (Frankle et al., 2020) extended the paths to the linear case. These insights enabled the transfer of the benefits regarding robustness of (traditional) output ensembles (Hansen & Salomon, 1990; Lakshminarayanan et al., 2017) to weight ensembles, reconciling the bias-variance trade-off (Belkin et al., 2019) while eliminating the computational cost of multiple inferences (Fort et al., 2020). These findings can be leveraged to improve performance on single-task (Izmailov et al., 2018; Wortsman et al., 2021; Ramé et al., 2022; Wortsman et al., 2022a; Jang et al., 2024), out-of-distribution (Wortsman et al., 2022b; Ramé et al., 2023), multi-task (Ilharco et al., 2022; Dimitriadis et al., 2023; 2025) and multi-objective alignment (Zhong et al., 2024; Ramé et al., 2024b) settings. Furthermore, model merging can also applied as a scalable approach to unify multiple task-specific models into a single model with multi-task capabilities (Ilharco et al., 2023; Yadav et al., 2023), despite performance loss compared to individual models. Several methods have tried to improve multi-task model merging by preserving the important parameters defined via the Fisher Information Matrix (Matena & Raffel, 2022; Tam et al., 2024), using heuristics (Davari & Belilovsky, 2023; Luo et al., 2023; Jin et al., 2023), randomly dropping and rescaling the task vector parameters (Yu et al., 2024) or by focusing on resolving weight interference caused by sign disagreements and redundant parameters (Yadav et al., 2023; Wang et al., 2024). Recent works use gradient descent to learn the layer-specific merging coefficients per task, e.g., Ada-merging (Yang et al., 2024) minimizes entropy in unlabeled test data while aTLAS (Zhang et al., 2024) optimizes using cross-entropy loss on validation data. Compared to `LiNeS`, these methods do not incorporate any prior knowledge on early vs. deep layers and require training, resulting in significant computational overheads.

## 3 POST-TRAINING LAYER-WISE SCALING MITIGATES FORGETTING

In this section, we present the key insight of our work: Scaling down the updates of shallow layers after fine-tuning can mitigate catastrophic forgetting and restore zero-shot generalization while preserving performance on the target task.

**Notation**    We consider a pre-trained model $\boldsymbol{\theta}_0 \in \mathbb{R}^N$ with $N$ parameters. Fine-tuning on a specific task $t$ results in the fine-tuned weights $\boldsymbol{\theta}_t$. The difference between these two sets of weights, $\boldsymbol{\tau}_t = \boldsymbol{\theta}_t - \boldsymbol{\theta}_0$, is referred to as the *task vector* or *residual* for task $t$ (Ilharco et al., 2023) and represents the updates made during fine-tuning.

**Fine-tuning leads to catastrophic forgetting**    We quantitatively demonstrate the phenomenon of catastrophic forgetting with the following experiments. Consider the 8-task image classification benchmark studied in Ilharco et al. (2023). We fine-tune a CLIP ViT-B/32 model on each task,

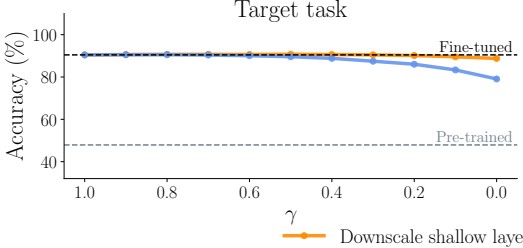 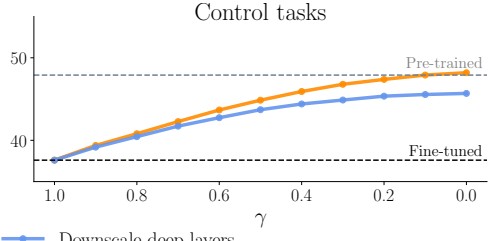

Figure 1: Downscaling the shallow layers maintains the fine-tuned performance on target tasks (orange line, left), while restoring zero-shot performance from pre-trained model on control tasks (orange line, right). The performance for downscaling deep layers instead is presented in blue lines, which underperforms downscaling shallow layers in both cases. $\gamma$ represents the minimum scaling factor applied to the layers, where a smaller $\gamma$ leads to stronger downscaling strength, with $\gamma = 1$ restoring the original fine-tuned model.

measuring performance on the fine-tuned task – referred to as the *target task* – and the remaining 7 tasks – the *control tasks*. The averaged results over all target and control task combinations, shown in Table 1, demonstrate that while fine-tuning significantly improves accuracy on the target task, it drastically reduces accuracy on the control tasks, underscoring the loss of the model's zero-shot generalization abilities.

**Shallow-layer updates impact minimally on target task accuracy**  Most parameter updates during the fine-tuning process are redundant, as similar performance is achievable without updating most pre-trained weights (Yadav et al., 2023; Wang et al., 2024; He et al., 2025). Moreover, prior work shows that task-specific features are often concentrated in deeper layers of the network (Neyshabur et al., 2020; Yosinski et al., 2014; Raghu et al., 2019). Based on these observations, we hypothesize that updates to the

Table 1: Fine-tuning harms generalization on control tasks. Our proposed post-training edition leads to a superior trade-off between performance on target and control tasks.

| Model / Accuracy | Target | Control |
|---|---|---|
| Pre-trained | 48.3 | 48.3 |
| Fine-tuned | 90.5 | 38.0 |
| Fine-tuned+LiNeS (ours) | 90.3 | 48.0 |

shallow layers contribute minimally to target tasks. To test this, we progressively downscale the updates to shallow layers after fine-tuning. Specifically, we apply a scaling factor to the updates to the $\ell$-th layer $\boldsymbol{\tau}^{(\ell)}$, defined as: $\lambda^{(\ell)} = \gamma + (1 - \gamma)\frac{\ell - 1}{L - 1}, \ \forall \ell \in [L], \gamma \in [0, 1]$, This linearly scales the updates from a factor of $\gamma$ for the first layer to 1 for the last one. As a result, fine-tuning updates to the shallow layers are scaled down more aggressively, with later layers experiencing progressively smaller reductions. We then reintroduce the scaled task vector into the pre-trained model and measure its performance on the fine-tuned task. Figure 1 (left) shows the results of this experiment for the CLIP ViT-B/32 checkpoint fine-tuned across the 8 tasks, where $\gamma$ is progressively decreased to strengthen the downscaling effect. We observe that, even with strong downscaling of shallow layers, the target task accuracy remains nearly unaffected. In contrast, when we downscale the deeper layers, target task accuracy drops significantly. These results support our hypothesis that shallow-layer updates are largely unnecessary for maintaining accuracy on the target task.

**Shallow-layer updates undermine zero-shot generalization**  While shallow-layer updates have minimal impact on target-task accuracy, they distort the general features learned during pre-training, which reside primarily in the shallow layers (Neyshabur et al., 2020; Yosinski et al., 2014; Raghu et al., 2019). We hypothesize that the degradation of performance on control tasks is largely due to these distortions in the shallow layers. Using the same experimental setup, we now evaluate the zero-shot performance on the control tasks, i.e., the other 7 unseen tasks. As shown in Figure 1 (right), as the strength of the shallow-layer downscaling increases, the accuracy on control tasks approaches the original pre-trained model's performance. This shows that by reducing the shallow-layer updates, we can restore most of the zero-shot performance that is lost during fine-tuning.

**Improved trade-off between target and control performance**  To optimize the trade-off between target and control task performance, we select a scaling coefficient $\gamma$ for each model that maximizes a weighted balance between these two objectives, as detailed in Appendix C.1. After selecting the optimal scaling coefficient, the test results are shown in the final row of Table 1. Our post-training method preserves target task accuracy with a minimal 0.2% difference while improving control task performance by 10%, compared to the fine-tuned model.

We further apply the same method to a 20-task computer vision benchmark (Wang et al., 2024). For evaluation, we report both the *target task normalized accuracy* and the *control task normalized accuracy* on the 19 tasks, where accuracy is normalized by the performance of the fine-tuned model for the target task and the

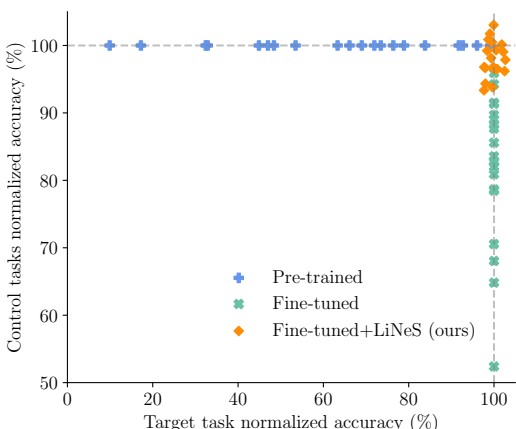

Figure 2: Our linear scaling (LiNeS) retains performance on both control and fine-tuned target tasks. Each dot represents a different model.

zero-shot accuracy of the pre-trained model for the control tasks. We compare to fine-tuned models on each task and the pre-trained model as baselines. Figure 2 shows that fine-tuning degrades zero-shot generalization, as indicated by the performance drop on control tasks. In contrast, our post-training scaling method significantly improves generalization while maintaining near-full target task accuracy. On average, our method achieves a target task normalized accuracy of 99.8% and a control task normalized accuracy 97.9%. This demonstrates its effectiveness in preserving both task-specific knowledge from fine-tuned checkpoints and the generalization capabilities of the pre-trained model. The full breakdown of results by task is available in Figure 6 in Appendix.

In Appendix, we show that catastrophic forgetting happens with models fine-tuned with LoRA (Hu et al., 2022) as well. As shown in Table 11, higher expressivity in the form of higher ranks increases target accuracy for LoRA but at the cost of lower performance on control tasks. Still, LiNeS significantly improves control performance while minimally affecting target accuracy. Furthermore, in Figure 10 of the appendix, we show that similar benefits can be observed for convolutional architectures such as ConvNeXt (Liu et al., 2022). Finally, we provide a performance comparison between editing models with LiNeS and regularized-fine-tuning-based methods in Appendix C.8, including applying different learning rates per layer. Also in these cases, LiNeS demonstrates superior performance on control tasks, while being much more computationally efficient.

## 4 METHOD

Motivated by the results of the previous section for mitigating forgetting, we propose LiNeS for **L**ayer-**i**ncreasing **Ne**twork **S**caling, a simple post-training technique that linearly rescales the updates of different layers in the task vector based on their depth in the network. LiNeS is designed to retain general features in the shallow layers while preserving the task-specific adaptations in the deeper layers.

Given a task vector $\boldsymbol{\tau}$ with $L$ layer blocks we apply the layer-wise linear scaling to adjust the contributions of shallow and deep layers using the following formulation:

$$\boldsymbol{\tau}_{\text{LiNeS}} = \texttt{concat}\left(\lambda^{(1)}\boldsymbol{\tau}^{(1)}, \ldots, \lambda^{(L)}\boldsymbol{\tau}^{(L)}\right), \quad \text{where } \lambda^{(\ell)} = \alpha + \beta\frac{\ell-1}{L-1}, \quad \forall \ell \in [L]. \quad (1)$$

As a result, the layers in $\boldsymbol{\tau}$ are progressively scaled with a factor between $\alpha$ for the first layer and $\alpha+\beta$ for the last layer, with intermediate layers scaled with a linearly increasing schedule depending on their depth. The final model $\boldsymbol{\theta}$ is then obtained by summing the pre-trained model weights and the edited task vector, i.e., $\boldsymbol{\theta} = \boldsymbol{\theta}_0 + \boldsymbol{\tau}_{\text{LiNeS}}$. Notice that, in Equation 1, $\boldsymbol{\tau}$ can correspond to either a single-task residual or, in the context of model merging, a multi-task vector obtained by merging the residuals of multiple checkpoints fine-tuned starting from a common initialization. Additional

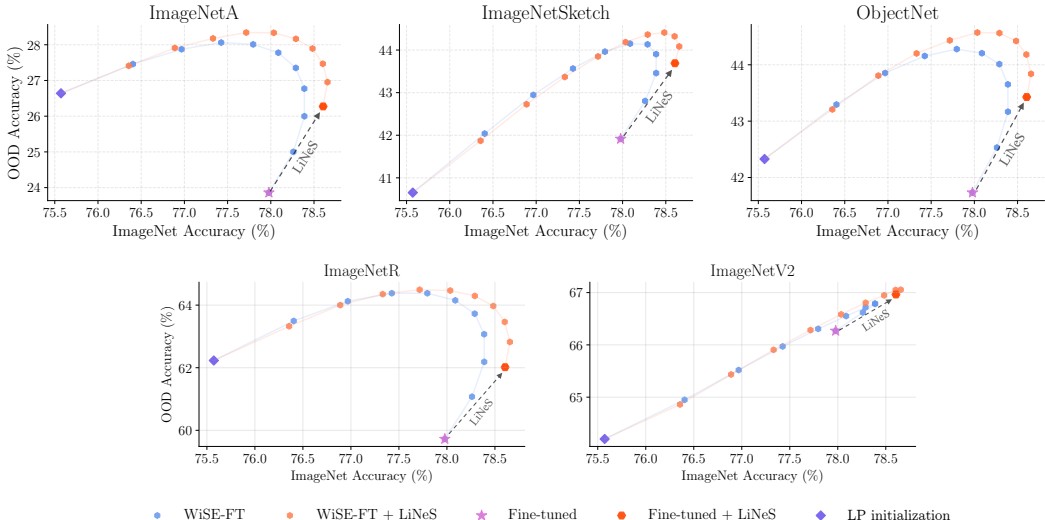

Figure 3: Application of `LiNeS` to WiSE-FT (Wortsman et al., 2022b) improves performance on ImageNet and five different distribution shifts, resulting in a dominating Pareto Front over WiSE-FT. LP initialization refers to the model initialized from a linear probe (Kumar et al., 2022).

details on this process are provided in the next section. Setting $\alpha = \beta = 0$ corresponds to the pre-trained model, while $\alpha = 1, \beta = 0$ is the fine-tuned model in the case that $\boldsymbol{\tau}$ is a single-task vector.

In practice, we find that tuning just one hyper-parameter (either $\alpha$ or $\beta$) is often sufficient to achieve a good balance between target task performance and generalization. Specific details on hyper-parameter tuning for different applications are provided in the experimental sections. The linear scaling method introduced in Section 3 corresponds to `LiNeS` by setting $\alpha = \gamma$ and $\beta = 1 - \gamma$. This formulation generalizes our previous approach, offering a flexible way to adjust the contributions of different layers based on the task requirements.

## 5  MODEL MERGING EXPERIMENTS

We empirically verify the effectiveness of applying `LiNeS` across diverse application domains. Section 5.1 presents results for improving robust fine-tuning (Wortsman et al., 2022b) for OOD generalization; Section 5.2 focuses on improving existing multi-task merging methods (Ilharco et al., 2023; Yadav et al., 2023; Wang et al., 2024) in both vision and NLP benchmarks. In Section 5.3, we apply `LiNeS` and improve the merging of single-task fine-tuned models within the setting of Model Soups (Wortsman et al., 2022a), and finally, we enhance merging foundation models fine-tuned on different rewards (Ramé et al., 2024a) in Section 5.4.

### 5.1  IMPROVING ROBUST FINE-TUNING FOR OOD GENERALIZATION

We first consider the setting of robust fine-tuning or WiSE-FT (Wortsman et al., 2022b), where linearly interpolating between the pre-trained and the fine-tuned weights improves model performance on OOD datasets. The interpolation is equivalent to scaling the residual $\boldsymbol{\tau}$: $(1-\gamma)\boldsymbol{\theta}_0 + \gamma\boldsymbol{\theta} = \boldsymbol{\theta}_0 + \gamma\boldsymbol{\tau}$, for $\gamma \in [0, 1]$. We apply `LiNeS` to the residual $\boldsymbol{\tau}$. Following Wortsman et al. (2022b), we evaluate CLIP models fine-tuned on ImageNet (Deng et al., 2009), considering 5 OOD datasets, namely ImageNetSketch (Wang et al., 2019), ImageNet-A (Hendrycks et al., 2021), ImageNet-R (Hendrycks et al., 2020), ObjectNet (Barbu et al., 2019), ImageNet-V2 (Recht et al., 2019).

We apply this `LiNeS` to each of the 70 fine-tuned checkpoints[2] provided by Wortsman et al. (2022a) setting $\alpha = \beta = 0.5$. We present the average results in Figure 3, comparing the performance of WiSE-FT with and without applying `LiNeS` on the 5 OOD datasets. Without applying WiSE-FT,

---

[2]The checkpoints are CLIP ViT-B/32 models fine-tuned on ImageNet with different hyper-parameters.

Table 2: Results for multi-task model merging in vision classification benchmarks of 8 tasks (Ilharco et al., 2023), 14 tasks, and 20 tasks (Wang et al., 2024) for different vision transformer architectures. Applying `LiNeS` improves baseline performance for all benchmark/architecture combinations.

| Method | with `LiNeS` | ViT-B/32 | | | ViT-L/14 | | |
|---|---|---|---|---|---|---|---|
| | | 8 tasks | 14 tasks | 20 tasks | 8 tasks | 14 tasks | 20 tasks |
| Zero-shot | | 48.3 | 57.3 | 56.1 | 64.8 | 68.3 | 65.3 |
| Fine-tuned | | 90.5 | 89.5 | 90.4 | 94.0 | 93.3 | 94.0 |
| Task Arithmetic | ✗ | 69.7 | 65.0 | 60.3 | 84.0 | 79.2 | 74.0 |
| | ✓ | **74.2** (+4.5) | **69.1** (+4.1) | **63.4** (+3.1) | **86.5** (+2.5) | **82.2** (+3.0) | **77.1** (+3.1) |
| Ties-Merging | ✗ | 73.6 | 67.6 | 63.1 | 85.6 | 79.3 | 75.6 |
| | ✓ | **77.2** (+3.6) | **72.1** (+4.5) | **67.2** (+4.1) | **88.0** (+2.4) | **82.5** (+3.2) | **79.6** (+4.0) |
| Consensus Merging | ✗ | 74.5 | 70.1 | 65.3 | 85.2 | 81.9 | 78.7 |
| | ✓ | **77.6** (+3.1) | **73.6** (+3.5) | **68.6** (+3.3) | **87.3** (+2.1) | **84.0** (+2.1) | **81.0** (+2.3) |

`LiNeS` already enhances both the ID and OOD performance of the fine-tuned models by a notable margin. Starting from this edited model and applying the WiSE-FT interpolation with the pre-trained weights leads to a Pareto Front (Caruana, 1997) that consistently dominates the one by WiSE-FT across all distribution shifts, illustrating the applicability of the proposed method across various distribution shifts. A granular result for applying `LiNeS` to each of the 70 checkpoints is provided in Appendix C.10.2, further highlighting its universal effectiveness across models. We also report similar findings in Figure 11 in Appendix for a CLIP ViT-B/16 checkpoint fine-tuned on ImageNet, using the same hyper-parameters as Wortsman et al. (2022b).

## 5.2 IMPROVING MULTI-TASK MODEL MERGING

In this section, we extend `LiNeS` to improve multi-task merging algorithms, aiming to combine multiple models fine-tuned independently on different tasks into a single model (Matena & Raffel, 2022; Ilharco et al., 2023; Ortiz-Jimenez et al., 2023; Yadav et al., 2023; Hazimeh et al., 2024). Task arithmetic (Ilharco et al., 2023) proposed to decouple the contributions of the pre-trained model and individual task vectors, first generating a multi-task vector $\tau_{\text{MTL}} = g(\tau_1, \ldots, \tau_T)$ with a merging function $g : \mathbb{R}^N \times \cdots \times \mathbb{R}^N \mapsto \mathbb{R}^N$, and then adding back to the pre-trained checkpoint with a scaling factor to construct a multi-task model $\theta = \theta_0 + \lambda \cdot \tau_{\text{MTL}}$. The scalar coefficient $\lambda$ is tuned using a held-out validation set. Recent works (Yadav et al., 2023; Wang et al., 2024) follow the same protocol while improving the merging function $g$ for retaining more task information. We refer to Appendix B.1 for a more detailed explanation of these methods.

However, significant performance loss occurs between the merged multi-task model and the original fine-tuned checkpoints. This performance decrease partially stems from interference (Yadav et al., 2023; Wang et al., 2024) among task vectors, where the contribution of one task negatively impacts performance on others, leading to overall degradation. Task interference is linked to catastrophic forgetting, as the individual task vectors lose a significant amount of generalization ability to other tasks after fine-tuning and merging them leads to interference among each other. Therefore, we can edit each task vector with `LiNeS` before merging to restore the generalization to other tasks, or for simplicity, edit directly the merged multi-task vector to preserve the shallow and general features that are beneficial across tasks.

We enhance the merging methods by applying `LiNeS` on the merged multi-task vector $\tau_{\text{MTL}}$. For the linear scaling schedule, we tune only $\beta$ and set $\alpha$ using a heuristic that adjusts based on both the number of merged models and the merging method. Specifically, for task arithmetic which aggregates the individual task vectors through a simple summation operation: $\tau_{\text{sum}} = \sum_{i=1}^{N_{\text{models}}} \tau_i$, we set $\alpha = 1/N_{\text{models}}$. For other merging strategies which result in $\tau_{\text{MTL}}$ with different magnitudes of norms, e.g., aggregation with summation leads to a norm $\times N_{\text{models}}$ larger compared to averaging, we further multiply by a scaling term $\|\tau_{\text{sum}}\| / \|\tau_{\text{MTL}}\|$ to normalize their norm to simple summation. Overall, we set the intercept $\alpha$ to:

$$\alpha = \frac{1}{N_{\text{models}}} \frac{\|\tau_{\text{sum}}\|}{\|\tau_{\text{MTL}}\|}, \text{ where } \tau_{\text{sum}} = \sum_{i=1}^{N_{\text{models}}} \tau_i \qquad (2)$$

Table 3: Results for multi-task model merging methods in three NLP benchmarks with T5-large model. `LiNeS` improves baseline performance across merging methods and benchmarks.

| Method | with `LiNeS` | T5-large (Lester et al., 2021) | | |
|---|---|---|---|---|
| | | 7 NLP tasks (Yadav et al., 2023) | 8 QA tasks (Zhou et al., 2022) | 11 NLP tasks (Wang et al., 2024) |
| Zero-shot | | 44.9 | 33.1 | 36.9 |
| Fine-tuned | | 85.9 | 80.7 | 78.7 |
| Task Arithmetic | ✗ | 71.9 | 63.8 | 63.6 |
| | ✓ | **76.4** (+4.5) | **67.6** (+3.8) | **66.2** (+2.6) |
| Ties-Merging | ✗ | 71.6 | 63.0 | 64.0 |
| | ✓ | **72.0** (+0.4) | **66.0** (+3.0) | **66.4** (+2.4) |
| Consensus Merging | ✗ | 73.5 | 68.6 | **67.5** |
| | ✓ | **75.4** (+1.9) | **69.3** (+0.7) | **67.5** (+0.0) |

Therefore, we only tune $\beta$ and search over the same range as the constant scaling $\lambda$ used by the aforementioned merging techniques. As a result, `LiNeS` shares the same computational requirements as the baseline merging methods; we provide more details for the hyper-parameters in Appendix B.2.1, as well as a sensitivity analysis to the hyper-parameters in Appendix C.6. Specifically, we consider various multi-task model merging baselines, namely Task Arithmetic (Ilharco et al., 2023), Ties-merging (Yadav et al., 2023), Consensus Merging (Wang et al., 2024), enhancing them with `LiNeS` and evaluate on both computer vision and NLP benchmarks.

### 5.2.1 COMPUTER VISION

We experiment with the 8-task image classification benchmark proposed by Ilharco et al. (2023), as well as the more challenging 14-task and 20-task benchmarks from Wang et al. (2024). Detailed descriptions of task composition appear in Appendix B.2.2. We also examine the efficacy of `LiNeS` across the model scale axis, studying three vision transformer (Dosovitskiy et al., 2021), namely ViT-B/32, ViT-B/16 and ViT-L/14, as CLIP visual encoders (Radford et al., 2021) .

Table 2 presents the results for ViT-B/32 and ViT-L/14, while Appendix C.4 contains the ViT-B/16 experiments. We observe that `LiNeS` provides a significant improvement to *all* baseline merging methods across *all* tested scenarios, regardless of model sizes and total number of tasks. For example, for the 8-task benchmark with ViT-B/32, `LiNeS` improves task arithmetic by 4.5%, Ties-merging by 3.6% and consensus merging by 3.1%. For the challenging 20-task benchmark with ViT-L/14, `LiNeS` leads to consistent and significant improvements, improving task arithmetic by 3.1%, Ties-merging by 4.0% and consensus merging by 2.3%. The detailed performance on individual tasks for each tested scenario is presented in Appendix C.11.

### 5.2.2 NATURAL LANGUAGE PROCESSING

We also evaluate the effectiveness of `LiNeS` in NLP domain, including a 7-task NLP benchmark (Yadav et al., 2023), an 8-task Question-Answering benchmark (Zhou et al., 2022), and their combined 11-task benchmark (Wang et al., 2024). Appendix B details the experimental settings. Following Tam et al. (2024), we adopt a variant of T5-large model (Raffel et al., 2020), namely T5-large-LM-Adapt (Lester et al., 2021), and use their provided checkpoints. While T5-large contains both encoder and decoder networks, we apply `LiNeS` only to the decoder, as our findings in Appendix C.5 indicate that applying the edition to the decoder leads to similar observations to vision.

The performance of applying `LiNeS` to baseline methods with T5-large across various NLP tasks is summarized in Table 3. `LiNeS` consistently improves multi-task performance across baseline merging methods and benchmarks with a notable margin. For example, on the 7 NLP tasks benchmark, `LiNeS` improves task arithmetic by 4.5 points, and consensus merging by 1.9 points. Meanwhile, `LiNeS` outperforms Ties-merging by 3.0% and 2.4% for the 8-QA benchmark and 11-NLP benchmark, respectively.

### 5.3 IMPROVING MODEL SOUPS FOR MERGING SINGLE-TASK MODELS

Averaging in weight space multiple models fine-tuned on the same task derived from the same pre-trained model has been shown to increase target performance (Wortsman et al., 2022a; Ramé

Table 4: `LiNeS` improves performance over Model Soups (Wortsman et al., 2022a), for both uniform and greedy soup in merging multiple checkpoints fine-tuned on ImageNet with different hyper-parameter configurations.

| Method | Enhancements | ImageNet Acc. |
|---|---|---|
| Averaged accuracy | / | 77.98 |
| Best individual model | / | 80.36 |
| Uniform soup | / | 79.99 |
| | Task Arithmetic | 80.17 |
| | `LiNeS` | **80.47** (+0.48) |
| Greedy soup | / | 81.01 |
| | Task Arithmetic | 81.01 |
| | `LiNeS` | **81.16** (+0.15) |

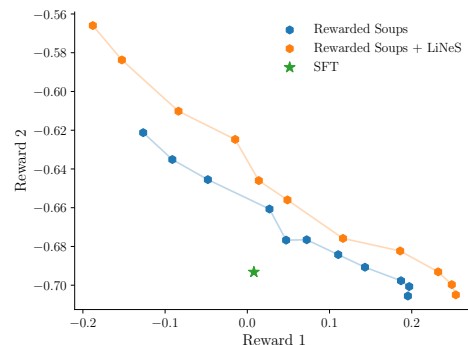

Figure 4: Applying `LiNeS` to Rewarded Soups (Ramé et al., 2023) improves merging of LLM policies RL fine-tuned on different rewards with a dominating Pareto Front.

et al., 2022). In this section, we investigate whether `LiNeS` can enhance the test performance when merging single-task models.

We follow the setting in Model Soups (Wortsman et al., 2022a) and merge 70 CLIP ViT-B/32 checkpoints fine-tuned on ImageNet (Deng et al., 2009) using different hyper-parameters, plus the pre-trained checkpoint. We consider both variants introduced in Wortsman et al. (2022a), namely uniform and greedy soup. We refer to Appendix B.3 for details regarding these methods and experimental settings. For both cases, the weight-averaging process can be decomposed as follows:

$$\boldsymbol{\theta}_{\text{soup}} = \boldsymbol{\theta}_0 + \boldsymbol{\tau}_{\text{soup}}, \text{ where } \boldsymbol{\tau}_{\text{soup}} = \frac{1}{N_{\text{models}}} \sum_{i=1}^{N_{\text{models}}} (\boldsymbol{\theta}_i - \boldsymbol{\theta}_0) \tag{3}$$

We apply `LiNeS` to $\boldsymbol{\tau}_{\text{soup}}$ fixing $\alpha = 1$ and searching over $\beta$. As a baseline, we also consider task arithmetic, where we search for a constant scaling factor on $\boldsymbol{\tau}_{\text{soup}}$. Note that both settings introduce one hyper-parameter to vanilla model soups, and refer to Appendix B.3 for a detailed description of the modifications. Table 4 summarizes the results and shows that `LiNeS` improves over vanilla soups and task arithmetic for both uniform and greedy soup by 0.48% and 0.15% on ImageNet, respectively. We report the best-performing model and the average performance as baselines. Finally, our proposed method compounds the gains from the greedy soup and leads to the best-performing model.

## 5.4 IMPROVING REWARDED SOUPS

In this section, we explore the effectiveness of `LiNeS` for merging foundation models fine-tuned on different rewards. We consider the Rewarded Soups setting (Ramé et al., 2023), which interpolates the weights $\boldsymbol{\theta}_1$ and $\boldsymbol{\theta}_2$ of two LLM policies, each optimized for a distinct reward $R_1$ and $R_2$, respectively.

Starting with an LLM parameterized by weights $\boldsymbol{\theta}_0$, we first fine-tune it using supervised fine-tuning (SFT) on labeled demonstrations. From the resulting weights $\boldsymbol{\theta}_{\text{SFT}}$, we then apply Reinforcement Learning from Human Feedback (RLHF) (Christiano et al., 2017; Ouyang et al., 2022), training two independent policies via Proximal Policy Optimization (PPO) (Schulman et al., 2017) to maximize the rewards $R_1$ and $R_2$ respectively. To merge these policies, we linearly interpolate the residuals $\boldsymbol{\tau}_1 = \boldsymbol{\theta}_1 - \boldsymbol{\theta}_{\text{SFT}}$ and $\boldsymbol{\tau}_2 = \boldsymbol{\theta}_2 - \boldsymbol{\theta}_{\text{SFT}}$, defining a continuous set of rewarded policies:

$$\boldsymbol{\theta}_{RS} = \boldsymbol{\theta}_{\text{SFT}} + \lambda \boldsymbol{\tau}_1 + (1 - \lambda) \boldsymbol{\tau}_2, \quad \lambda \in [0, 1], \tag{4}$$

where the coefficient $\lambda$ models the user's preferences. We apply `LiNeS` to the weighted-sum residual: $\lambda \boldsymbol{\tau}_1 + (1 - \lambda) \boldsymbol{\tau}_2$, fixing $\alpha = \beta = 1$ for computational reasons.

In our experiment, we use LLaMA-2 7B (Touvron et al., 2023) and the Reddit Summary task (Stiennon et al., 2020), which consists of 14.9k post-summary pairs. We fine-tune the model using LoRA (Hu et al., 2022) with $r_{\text{LoRA}} = 64$, $\alpha_{\text{LoRA}} = 128$, and 0.05 dropout. We employ two reward models:

GPT2-reward-summarization – which scores summaries based on human preferences – and BART-faithful-summary-detector (Chen et al., 2021) – which evaluates the faithfulness of the generated summary to the source post. To evaluate the models, we use a subset of 1k samples from the test set, generate the responses, and compute the average score for each reward dimension.

In Table 4, we present the empirical Pareto Fronts for both Rewarded Soups and Rewarded Soups+`LiNeS`. `LiNeS` consistently outperforms the vanilla Rewarded Soups across the full preference space, Pareto dominating the baseline. This result highlights the generality of `LiNeS`.

## 6 DISCUSSION

We compare `LiNeS` with prior work that optimizes the scaling coefficients via backpropagation. Specifically, Ada-merging (Yang et al., 2024) minimizes the entropy loss of the predictions on the test set, while aTLAS (Zhang et al., 2024) minimizes a cross entropy loss on validation samples. Both methods operate on a more fine-grained level and introduce coefficients per layer *and* per task, requiring all $T + 1$ checkpoints, for task vectors and pre-trained model respectively, to be stored in memory during their fine-tuning process.

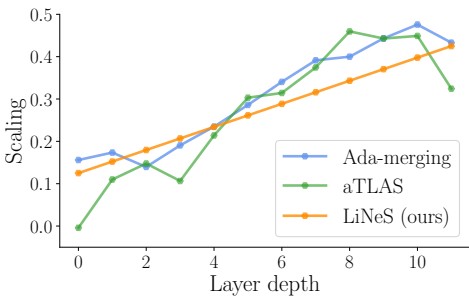

Figure 5: Comparison of the scalings obtained by different methods on 8-task merging benchmark with CLIP ViT-B/32.

We consider the 8-task computer vision benchmark with the ViT-B/32 visual encoder, and present the per-layer scalings in Figure 5. For aTLAS and Ada-merging, we report the average optimized scaling coefficients for attention and linear layers in each block across tasks. Without requiring training, `LiNeS` leverages the inductive bias of neural networks to achieve scaling very close to Ada-merging or aTLAS , but with much less computational cost. Apart from the excessive memory overhead, both aTLAS and Ada-merging require multiple training epochs, making it challenging to scale for large models. As we demonstrate in Section 5.4, `LiNeS` efficiently scales to large models like LLaMA (Touvron et al., 2023).

## 7 CONCLUSION

In this work, we presented `LiNeS`, a novel method designed to mitigate catastrophic forgetting after fine-tuning process. By reducing the magnitude for parameter updates in the shallower layers, `LiNeS` improves the generalization performance of the edited model on control tasks while almost fully preserving performance on the fine-tuned tasks. Furthermore, we demonstrated the versatility of `LiNeS` in addressing task interference in multi-task model merging, where it consistently improves the baseline model merging methods across vision and NLP benchmarks. Our experiments confirm the broad applicability of `LiNeS` across various scenarios, from improving OOD generalization to enhancing multi-task and single-task model merging strategies, as well as improving merging LLM policies aligned with different rewards. Given its simplicity and ease of integration with existing methods, `LiNeS` offers a practical and inexpensive solution for boosting the generalization and robustness of fine-tuned models in diverse application domains.

### ACKNOWLEDGMENTS

The authors thank Adam Hazimeh and the anonymous reviewers for their constructive discussions and comments.

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

# Appendix

## Table of Contents

# A  LINES PSEUDOCODE

We provide here a python pseudocode for the scaling the task vectors.

```python
def line_scaling(task_vector, alpha=0.0, beta=1.0, num_blocks=12):
    """
    Progressively scales the task vector based on layer depth.

    Parameters:
    -----------
    task_vector : dict
        A dictionaryves control performantween the fine-tuned
        checkpoint and the pre-trained checkpoint.
    alpha : float
         The minimum scaling factor for the blocks.
    beta : float
        The maximum scaling coefficient difference between the last and first
        block.
    num_blocks : int
        The total number of layer blocks in the model.
    Returns:
    --------
    scaled_task_vector : dict
        A copy of `task_vector` where each key is scaled based on the layer
        depth.
    """

    import copy

    # Deep copy the task vector to avoid modifying the original
    scaled_task_vector = copy.deepcopy(task_vector)

    # Generate the key blocks corresponding to the layers of the model
    key_blocks = [f".layer{i}." for i in range(num_blocks)]

    # Create a scaling dictionary to store the scaling factor for each key
    scaling_dic = {}
    for k in task_vector.keys():
        # Find the layer block in the key and assign scaling factor based
        # on layer depth
        for layer, block in enumerate(key_blocks):
            if block in k:
                scaling_dic[k] = alpha + beta * (layer / (num_blocks - 1))
                break

    # Scale the task vector based on the scaling dictionary
    scaled_task_vector.vector = {
        # Use alpha if layer is outside residual blocks
        k: task_vector.vector[k] * scaling_dic.get(k, alpha)
        for k in task_vector.keys()
    }

    return scaled_task_vector

# example: scale single-task fine-tuned residual
task_vector = {k: theta_t[k] - theta_0[k] for k in theta_0.keys()}
scaled_task_vector = line_scaling(
    task_vector, alpha=gamma, beta=1.0 - gamma, num_blocks=12
)

# example: Scale the multi-task vectors
mtv = {
    k: sum(theta_ft[k] - theta_0[k] for theta_ft in ft_models)
    for k in theta_0.keys()
}
scaled_mtv = line_scaling(
    mtv, alpha=1 / len(ft_models), beta=beta, num_blocks=12
)
```

# B  EXPERIMENTAL DETAILS

## B.1  DESCRIPTIONS OF BASELINE MODEL MERGING METHODS

- **Task Arithmetic** (Ilharco et al., 2023) generates a multi-task vector by summing the individual task vectors for each task. This multi-task vector is then added to the pre-trained checkpoint, with a scaling factor chosen based on validation set performance.
- **Ties-Merging** (Yadav et al., 2023) resolves parameter conflicts during model merging by first pruning parameters with lower magnitudes from the individual task vectors, followed by addressing sign mismatches, and finally merging parameters with consistent signs with averaging operation. The resulting multi-task vector is then added to the pre-trained checkpoint using a scaling factor determined from the validation set.
- **Consensus Merging** (Wang et al., 2024) enhances existing model merging techniques by eliminating redundant weights in the multi-task vector. It first identifies the relevant subset of parameters for each task, then filters out weights that are relevant to either none or only one task. After removing these redundant weights, the refined multi-task vector is added to the pre-trained checkpoint with a scaling factor selected from the validation set.

While consensus merging can be applied to various merging methods, in all our experiments, we evaluate only its application to task arithmetic.

## B.2  EXPERIMENTAL DETAILS FOR MULTI-TASK MODEL MERGING

### B.2.1  HYPER-PARAMETERS TUNING

We list here the hyper-parameter search space for each model merging method in Table B.2.1, while we suggest the authors to the original papers for a detailed description of these hyper-parameters. We highlight that applying `LiNeS` does not introduce extra computational cost in hyper-parameter search for the baseline merging methods.

| Method | With `LiNeS` | Hyper-parameter search space |
|---|---|---|
| Task Arithmetic | ✗ | constant scaling term for multi-task vector: [0.1,0.2,0.3,0.4,0.5,0.6,0.7,0.8,0.9,1.0] |
| | ✓ | scaling term $\beta$ in Eq. 1 for the multi-task vector: [0.1,0.2,0.3,0.4,0.5,0.6,0.7,0.8,0.9,1.0] |
| Ties-Merging | ✗ | constant scaling term for multi-task vector: [0.1,0.2,0.3,0.4,0.5,0.6,0.7,0.8,0.9,1.0,1.1,1.2,1.3,1.4,1.5] |
| | ✓ | scaling term $\beta$ in Eq. 1 for the multi-task vector: [0.1,0.2,0.3,0.4,0.5,0.6,0.7,0.8,0.9,1.0,1.1,1.2,1.3,1.4,1.5] |
| Consensus Merging | ✗ | constant scaling term for multi-task vector: [0.1,0.2,0.3,0.4,0.5,0.6,0.7,0.8,0.9,1.0]; weight-pruning threshold: [1, 2] |
| | ✓ | scaling term $\beta$ in Eq. 1 for the multi-task vector: [0.1,0.2,0.3,0.4,0.5,0.6,0.7,0.8,0.9,1.0]; weight-pruning threshold: [1, 2] |

### B.2.2  BENCHMARKS

**Image classification**  For the benchmarks used in image classification, we utilized the 8-task benchmark initially proposed by Ilharco et al. (2023), as well as the 14-task and 20-task benchmarks expanded by Wang et al. (2024).

- The **8-task benchmark** comprises the following tasks: Cars (Krause et al., 2013), DTD (Cimpoi et al., 2014), EuroSAT (Helber et al., 2019), GTSRB (Stallkamp et al., 2011), MNIST (LeCun, 1998), RESISC45 (Cheng et al., 2017), SUN397 (Xiao et al., 2016), and SVHN (Netzer et al., 2011).
- The **14-task benchmark** includes the original eight tasks plus additional ones: CIFAR100 (Krizhevsky & Hinton, 2009), STL10 (Coates et al., 2011), Flowers102 (Nilsback & Zisserman, 2008), OxfordIIITPet (Parkhi et al., 2012), PCAM (Veeling et al., 2018), and FER2013 (Goodfellow et al., 2013).
- The **20-task benchmark** builds on the 14-task benchmark with the addition of: EMNIST (Cohen et al., 2017), CIFAR10 (Krizhevsky & Hinton, 2009), Food101 (Bossard et al., 2014), FashionMNIST (Xiao et al., 2017), RenderedSST2 (Socher et al., 2013; Radford et al., 2019), and KMNIST (Clanuwat et al., 2018).

**Natural Language Processing** For our NLP experiments, we utilized benchmarks established by Yadav et al. (2023), Tam et al. (2024), and Wang et al. (2024).

- The **7 NLP Tasks** benchmark, as explored in Yadav et al. (2023), includes the following datasets: QASC (Khot et al., 2020), QuaRTz (Tafjord et al., 2019), PAWS (Zhang et al., 2019), Story Cloze (Sharma et al., 2018), WikiQA (Yang et al., 2015), Winogrande (Sakaguchi et al., 2021), and WSC (Levesque et al., 2012).
- The **8 QA Tasks** (Tam et al., 2024) comprises the following datasets: CosmosQA Huang et al. (2019), QASC (Khot et al., 2020), QuAIL Rogers et al. (2020), QuaRTz (Tafjord et al., 2019), PAWS (Zhang et al., 2019), ROPES Lin et al. (2019), SocialIQA Sap et al. (2019), and WikiQA (Yang et al., 2015).
- The **11 NLP Tasks** benchmark is a union of these two benchmarks, as studied in Wang et al. (2024). It contains the following tasks: QASC (Khot et al., 2020), QuaRTz (Tafjord et al., 2019), PAWS (Zhang et al., 2019), Story Cloze (Sharma et al., 2018), WikiQA (Yang et al., 2015), Winogrande (Sakaguchi et al., 2021), WSC (Levesque et al., 2012), CosmosQA Huang et al. (2019), QuAIL Rogers et al. (2020), ROPES Lin et al. (2019), and SocialIQA Sap et al. (2019).

### B.3 EXPERIMENTAL DETAILS FOR SINGLE-TASK MODEL MERGING

#### B.3.1 DESCRIPTION OF MODEL SOUPS

**Model soups** (Wortsman et al., 2022a) is a model merging method which averages the weights of multiple fine-tuned models with different hyper-parameter configurations, improving accuracy of the merged model without increasing inference or memory costs. The authors of model soups proposed two methods:

- **Uniform soup**: Averages the weights of all fine-tuned checkpoints, providing a simple and efficient way to improve performance.
- **Greedy soup**: Starting with the best-performing checkpoint, greedily and iteratively adds the next best-performing checkpoint to the soup, keeping those that improve accuracy of current collection of model checkpoints.

#### B.3.2 EXPERIMENTAL DETAILS FOR MODIFICATIONS TO MODEL SOUPS

We describe in detail the modifications to model soups, namely task arithmetic and our proposed `LiNeS`. For reference, model soups merges the checkpoints by averaging the weights of the individual checkpoints:

$$\boldsymbol{\theta}_{\text{soup}}^{\text{vanilla}} = \boldsymbol{\theta}_0 + \boldsymbol{\tau}_{\text{soup}} \tag{5}$$

**Enhancing Model Soups with Task Arithmetic** We enhance model soups with task arithmetic, by introducing a scaling factor $\lambda_{\text{ta}}$ to $\boldsymbol{\tau}_{\text{soup}}$ in Equation 5. We search for this hyper-parameter within the range of $[1.0, 1.1, 1.2, 1.3, 1.4, 1.5, 1.6, 1.7, 1.8, 1.9, 2.0]$. Note that $\lambda_{\text{ta}} = 1.0$ yields the vanilla model soups.

$$\boldsymbol{\theta}_{\text{soup}}^{\text{ta}} = \boldsymbol{\theta}_0 + \lambda_{\text{ta}} \cdot \boldsymbol{\tau}_{\text{soup}} \tag{6}$$

**Enhancing Model Soups with `LiNeS`** We enhance model soups with `LiNeS`, by applying `LiNeS` to $\boldsymbol{\tau}_{\text{soup}}$ in Equation 5. For the scaling, we apply directly the scaling introduced in Equation 1 to create a scaled task vector $\boldsymbol{\tau}_{\text{soup}}^{\text{LiNeS}}$, fixing $\alpha$ to 1 while searching the value for $\beta$ within the range of $[0.0, 0.1, 0.2, 0.3, 0.4, 0.5, 0.6, 0.7, 0.8, 0.9, 1.0]$. Note that $\beta = 0.0$ yields the vanilla model soups.

$$\boldsymbol{\theta}_{\text{soup}}^{\text{LiNeS}} = \boldsymbol{\theta}_0 + \boldsymbol{\tau}_{\text{soup}}^{\text{LiNeS}} \tag{7}$$

We further note that, both Task Arithmetic and our proposed method introduce only one hyper-parameter to model soups, while the computational cost for hyper-parameter search is the same.

When applying the modifications to greedy soup, we only apply them directly to the selected subset of checkpoints after the greedy selection process.

We search for the hyper-parameter within the validation set and report the performance on test set with the best hyper-parameter based on validation performance.

## C  ADDITIONAL RESULTS

### C.1  DIFFERENT TRADE-OFFS FOR RETENTION OF TASK AND CONTROL TASK PERFORMANCE

In Section 3, we need to balance two competing objectives: maximizing accuracy on the target task while preserving performance on the control task. To account for different user preferences, we scalarize these objectives by assigning varying weights to the target task accuracy. This weighting scheme can be adjusted depending on the scenario to reflect different priorities. Let $w_{\text{target}}$ represent the weight assigned to the target task accuracy, and $M_{\text{target}}$ and $M_{\text{control}}$ denote the normalized accuracies for the target and control tasks, respectively. The optimal value of $\gamma$ is selected to maximize the following weighted trade-off on the validation set:

$$w_{\text{target}}M_{\text{target}} + M_{\text{control}}$$

To account for the high variance in control task performance and to emphasize the target task, we assign it a weight of 2, signifying that its accuracy is prioritized twice as much as the control task's accuracy.

Table 5: Validation results on the target vs control performance benchmark, presented in Section 3, averaged over the 8 tasks. We balance two competing objectives with various scalarization weights $w_{\text{target}}$. In the main text, we use $w_{\text{target}} = 2$.

| $w_{\text{target}}$ | Averaged normalized accuracy (%) | |
| --- | --- | --- |
| | Target task | Control tasks |
| 1 | 99.8 | 101.9 |
| 2 | 100.0 | 101.5 |
| 5 | 100.2 | 100.8 |

### C.2  DETAILED LABELS FOR FIGURE 2

We provide in Figure 6 the detailed labels corresponding to each scatter dot. Each scatter dot corresponds to applying a specific model (FT for fine-tuned model; PT for pre-trained model; LS for fine-tuned model edited with `LiNeS`) on different tasks.

### C.3  ABLATIONS OF DIFFERENT CHOICES OF SCALING FUNCTION

We provide in this section an ablation study on applying different scaling functions for `LiNeS`. In `LiNeS` we used directly $\lambda^{(\ell)} = \alpha + \beta \cdot \frac{\ell-1}{L-1}$ to scale different layers. Here we test the performance on multi-task model merging in vision benchmarks with the following choices for scaling functions $f(\cdot)$: linear scaling, quadratic scaling and square root scaling:

- linear scaling: $\lambda^{(\ell)} = \alpha + \beta \cdot \frac{\ell-1}{L-1}$

- square root scaling: $\lambda^{(\ell)} = \alpha + \beta \cdot \left(\frac{\ell-1}{L-1}\right)^{\frac{1}{2}}$

- quadratic scaling: $\lambda^{(\ell)} = \alpha + \beta \cdot \left(\frac{\ell-1}{L-1}\right)^{2}$

We provide in Table C.3 the performance of different choices of scaling on vision benchmarks with ViT-B/32. While using quadratic scaling sometimes outperforms using identify function, especially with a larger number of tasks during merging, the improvement is not substantial. Therefore, we choose the linear scaling to keep the method simple and general.

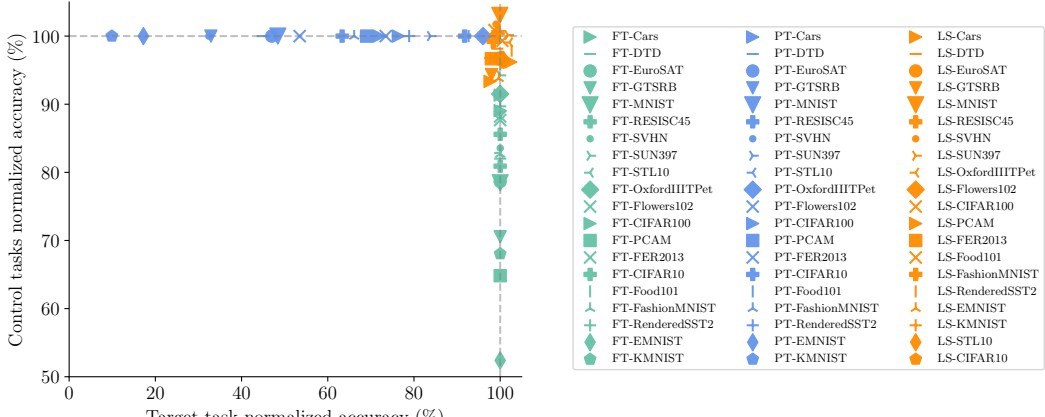

Figure 6: Figure 2 with detailed label information. Each scatter dot corresponds to applying a specific model (FT for fine-tuned model; PT for pre-trained model; LS for fine-tuned model edited with `LiNeS`) on different task.

Table 6: Ablation study for applying different scalings for `LiNeS` on vision benchmarks with ViT-B/32.

| Method | Scaling function | ViT-B/32 | | |
|---|---|---|---|---|
| | | 8 tasks | 14 tasks | 20 tasks |
| Task Arithmetic | linear | **74.2** | 69.1 | 63.4 |
| | square root | 73.9 | 67.5 | 62.4 |
| | quadratic | 73.8 | **69.2** | **64.6** |
| Ties-Merging | linear | **77.2** | **72.1** | 67.2 |
| | square root | 76.9 | 70.4 | 65.6 |
| | quadratic | 76.1 | 71.6 | **67.4** |
| Consensus Merging | linear | **77.6** | 73.6 | 68.6 |
| | square root | 77.1 | 72.5 | 67.3 |
| | quadratic | 77.1 | **73.9** | **69.0** |

## C.4 RESULTS FOR VIT-B/16 FOR MULTI-TASK MERGING

We provide in Table C.4 the results complementary to Table 2 for using ViT-B/16 as the image encoder, where we observe similar performance gains and observations by using `LiNeS` as in Table 2.

Table 7: Complementary to Table 2, for results obtained with ViT-B/16 as image encoder.

| Method | with `LiNeS` | ViT-B/16 | | |
|---|---|---|---|---|
| | | 8 tasks | 14 tasks | 20 tasks |
| Zero-shot | | 55.5 | 61.4 | 59.8 |
| Fine-tuned | | 92.6 | 91.6 | 92.3 |
| Task Arithmetic | ✗ | 74.6 | 70.4 | 65.7 |
| | ✓ | **77.6** (+3.0) | **72.7** (+2.3) | **67.7** (+2.0) |
| Ties-Merging | ✗ | 79.1 | 73 | 68.1 |
| | ✓ | **79.9** (+0.8) | **75.2** (+2.2) | **71.2** (+3.1) |
| Consensus Merging | ✗ | 78.9 | 73.9 | 70.2 |
| | ✓ | **79.5** (+0.6) | **75.8** (+1.9) | **72.0** (+1.8) |

## C.5 Results for editing T5 with LiNeS

We repeat here similar experiments we performed on ViT-B/32 model in Section 3 with T5-large (Raffel et al., 2020). T5-large contains both encoder and decoder structure, with sequential residual blocks in both structures. We investigate separately how the shallow-layer updates in the encoder and decoder infect the target and control task accuracy.

We consider the 8-question-answering benchmark (Zhou et al., 2022), and plot in Figure 7 the averaged target and control task accuracy after applying LiNeS to

1. only the decoder part (left),
2. only the encoder part (middle),
3. both the encoder and decoder part (right).

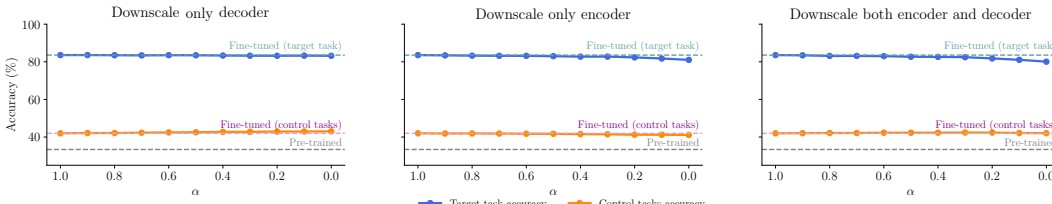

Figure 7: The impact of downscaling the shallower-layer parameter updates on T5-large model within the 8-question-answering benchmark (Raffel et al., 2020). Downscaling only on the decoder (left) architecture preserves full target performance, while slightly improving the control tasks performance. Downscaling on the encoder leads to performance degradation on target tasks.

We observe that, only downscaling on the decoder architecture fully preserves full target performance, while downscaling on the encoder, or on both encoder and decoder, leads to performance drop on the target tasks. On the other hand, downscaling on the decoder part slight improves control generalization of the fine-tuned model, which we do not observe from downscaling on the encoder, or simultaneously on the encoder and decoder. We also note that, unlike the case in vision, the fine-tuned checkpoints on this NLP benchmark actually improve over the zero-shot performance of the pre-trained model on control tasks.

These results motivate us to apply LiNeS to only the decoder part of T5-large when merging multiple checkpoints, which preserves full target task accuracy while slightly improving control task performance, leading to similar observation in applying LiNeS to the ViT-B/32 architecture in vision,

## C.6 Sensitivity analysis for hyper-parameters

We provide in this section the sensitivity analysis for the hyper-parameters of LiNeS. Specifically, we consider the setting in multi-task merging in the 8-task vision classification benchmark with ViT-B/32 CLIP model.

As explained in Section 5.2, LiNeS fixes $\alpha$ with a heuristic value by Equation 2 and only tunes $\beta$ for multi-task merging. The slope hyper-parameter $\beta$ is tuned within the same range as the uniform scaling coefficient $\lambda$ for the baseline merging methods. We compare in Figure 8 the sensitivity of averaged multi-task validation accuracy to the respective hyper-parameters, i.e., to $\lambda$ for baseline merging methods and $\beta$ for the LiNeS-enhanced merging methods. The results show that, for all three merging methods, including Task arithmetic, Ties-merging and Consensus, enhancing with LiNeS is less sensitive to hyper-parameter choices compared to the corresponding baseline method.

Furthermore, we perform an ablation treating $\alpha$ as a hyper-parameter and analyze the sensitivity to both $\alpha$ and $\beta$ for LiNeS in a two-dimensional grid for the same benchmark. The results are presented in Figure 9. The results clearly demonstrate the necessity for applying layer-increasing scaling, as the optimal performance is obtained with both $\alpha > 0$ and $\beta > 0$ for all three merging method. Note that the optimal configurations found by the ablation study are very close to the configurations found in our method, as shown in Table 8, by setting $\alpha$ via the heuristic and searching only for $\beta$.

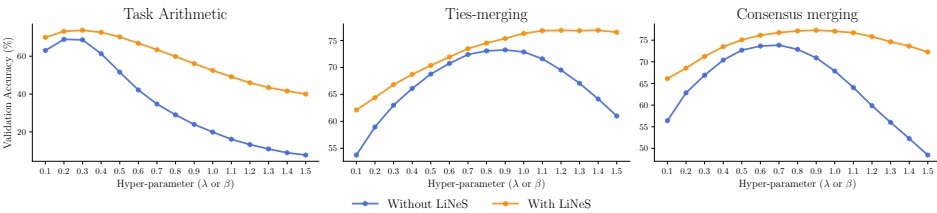

Figure 8: Sensitivity to hyper-parameters in multi-task merging for the 8-task benchmark with CLIP ViT-B/32 model. The y-axis represents the averaged multi-task validation accuracy and x-axis represents the hyper-parameter value, i.e., $\lambda$ for the baseline method and $\beta$ for the method enhanced with `LiNeS`.

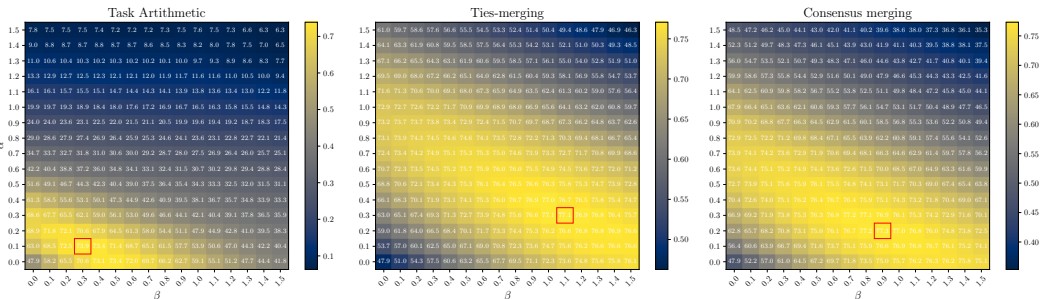

Figure 9: Sensitivity to both $\alpha$ and $\beta$ in multi-task merging for the 8-task benchmark with CLIP ViT-B/32 model. The heatmap represents the averaged multi-task validation accuracy, while x and y axis represent the $\beta$ and $\alpha$ respectively. The optimal configuration is annotated with a red box.

## C.7 EXPERIMENTS WITH CNN ARCHITECTURES

In this section, we apply `LiNeS` to CNN architectures. Specifically, we consider the ConvNeXt (Liu et al., 2022) architecture. First, we repeat the experiments presented in Section 3 regarding mitigating catastrophic forgetting. The final results are presented in Figure 10, where we observe similar findings with CLIP ViTs, i.e., `LiNeS` greatly improves the performance on control tasks when applied to the fine-tuned checkpoints while preserving most of the accuracy on target tasks. Furthermore, we present in Table 9 the results on multi-task model merging, following the experimental protocol established in Section 5.2. Again, we see that `LiNeS` improves the performance of baseline merging methods.

## C.8 EXPERIMENTS WITH REGULARIZED FINE-TUNING

In this section, we evaluate `LiNeS` against several regularized fine-tuning methods, focusing on their ability to preserve general features and mitigate catastrophic forgetting. The regularization strategies applied during fine-tuning are described below:

1. **Fine-tuning with Linear Layer-Wise Learning Rate Decay (LinLR):** Applies a linear learning rate schedule where the learning rate linearly increases from 0.0 to the maximum value for all the layers.

2. **Fine-tuning with Exponential Layer-Wise Learning Rate Decay (ExpLR):** Applies an exponential learning rate schedule where the learning rate is set to maximum for the deepest layers and decays by a factor of 0.5 by each layer for the shallower layers.

3. **Fine-tuning with First Half of Blocks Frozen (HalfFT):** Freezes the parameters of the first half of the model's blocks during training.

4. **Fine-tuning only the Final Block (LastFT):** Freezes all blocks except the final block of the feature encoder.

Table 8: $\alpha$ and $\beta$ values for $\alpha$ set by our proposed heuristic in Equation 2.

| Method | $\alpha$ | $\beta$ |
|---|---|---|
| Task Arithmetic | 0.125 | 0.3 |
| Ties-merging | 0.21 | 1.4 |
| Consensus merging | 0.21 | 0.9 |

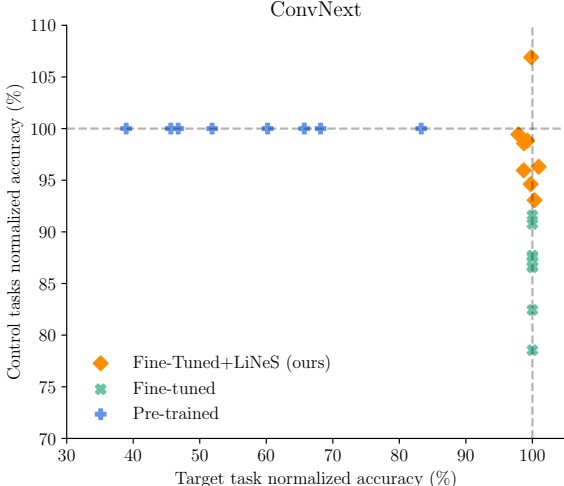

Figure 10: Our linear scaling (LiNeS) retains performance on both control and fine-tuned target tasks for ConvNext architecture.

We present the results in Table 10. We observe that LiNeS, as a post-training editing method, outperforms the regularized fine-tuning methods in terms of restoring the zero-shot performance on the control tasks. We further emphasize that compared with the regularized fine-tuning methods, LiNeS benefits from many advantages such as efficiency, flexibility and computational cost.

## C.9 EXPERIMENTS WITH LoRA FINE-TUNING

We explore the applicability of the method on models fine-tuned with LoRA (Hu et al., 2022). For a layer with pre-trained weights $W_0 \in \mathbb{R}^{m \times n}$, LoRA adds trainable matrices $A \in \mathbb{R}^{m \times r}$ and $B \in \mathbb{R}^n$, for rank $r \ll \min(n, m)$. The weights of the layer become:

$$W = W_0 + \frac{\alpha}{r} BA$$

for $\alpha \in \mathbb{R}$. Following common practice, we set $\alpha = r$ and fine-tune with the same protocol used for full fine-tuning. We consider only the case of ViT-B/32 fine-tuned on 8 tasks and replicate the experiment presented in Table 1. Specifically, for each of the 8 LoRA-fine-tuned models, we compute the accuracy on the same (target) task as well as the average performance for each of the 7 remaining control tasks. Table 11 reports the average over the 8 cases for ranks $r \in \{16, 32, 64, 128\}$. We observe that LoRA fine-tuning has lower target performance compared to full fine-tuning and that increasing target performance comes at the cost of more forgetting. In all the cases, LiNeS restores control performance while minimally affecting target performance.

## C.10 ADDITIONAL RESULTS FOR IMPROVING WISE-FT WITH LiNeS

### C.10.1 RESULTS FOR USING ViT-B/16 AS VISUAL ENCODER

We provide in Figure 11 the results for applying LiNeS for improving WiSE-FT, using ViT-B/16 as visual encoder. The ViT-B/16 checkpoint obtained through fine-tuning the CLIP checkpoint on ImageNet with the same hyper-parameter configurations in Wortsman et al. (2022b). From Figure 11

Table 9: Multi-task model merging results using a ConvNeXt architecture. LiNeS improves the results compared to uniform scaling for both Task Arithmetic and Ties-merging.

| Method | LiNeS | Acc (%) | Norm. Acc (%) |
|---|---|---|---|
| Task Arithmetic | ✗ | 77.9 | 83.8 |
| | ✓ | 79.0 [+1.1] | 84.8 [+1.0] |
| Ties-merging | ✗ | 79.7 | 85.8 |
| | ✓ | 80.3 [+0.6] | 86.3 [+0.5] |

Table 10: Performance of different methods on target and control tasks, averaged over all target and control task combinations in the 8-task vision benchmark (Ilharco et al., 2023).

| | pre-trained | fine-tuned | FT+LiNeS | FT+LinLR | FT+ExpLR | FT+HalfFT | FT+LastFT |
|---|---|---|---|---|---|---|---|
| Target (%) | 48.3 | 90.5 | 90.3 | 90.7 | 89.6 | 90.4 | 85.6 |
| Control (%) | 48.3 | 38.0 | 48.0 | 46.9 | 46.0 | 46.8 | 46.6 |

we observe that LiNeS improves over WiSE-FT for both ID and OOD accuracies, leading to similar observations as the results obtained with ViT-B/32.

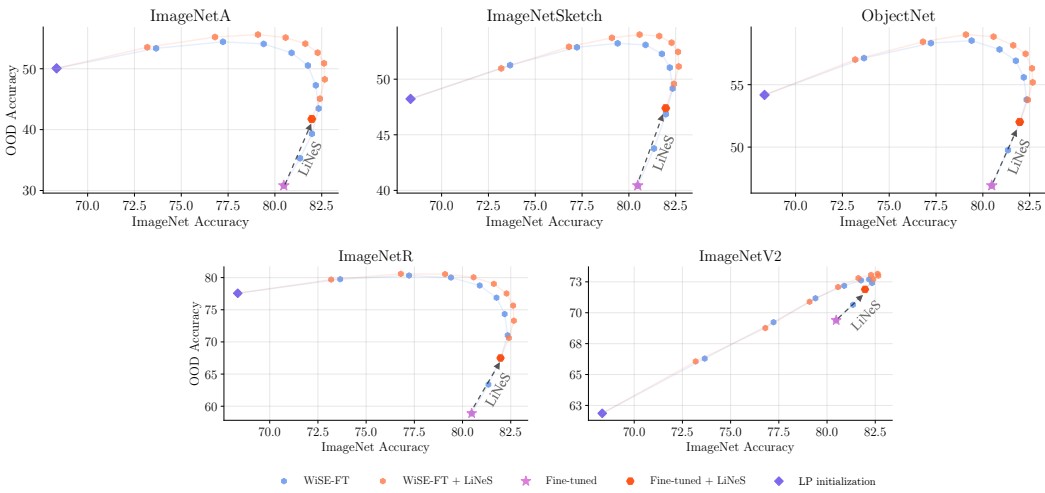

Figure 11: Results for improving WiSE-FT with LiNeS on with ViT-B/16 model fine-tuned on ImageNet.

### C.10.2 INDIVIDUAL RESULTS FOR 70 CHECKPOINTS

We provide in Figure 12 individual results separately for the experiments on the 70 individual model checkpoints. Note that here y-axis represents the averaged accuracy over 5 OOD datasets. From the figure, we observe that LiNeS consistently improves WiSE-FT in terms of both ID and OOD accuracies for most of the individual checkpoints.

### C.11 DETAILED PERFORMANCE ON INDIVIDUAL TASKS FOR MULTI-TASK MODEL MERGING

**Image Classification** We provide the detailed performance on each individual task for multi-task model merging in image classification benchmarks, complementary to the results in Table 2 and Table C.4 where the accuracies are averaged on the individual tasks.

Table 11: Similar to Table 1, LoRA Fine-tuning harms generalization on control tasks. Increased target performance rsults in higher levels of forgetting. Still, our proposed method LiNeS restores control performance for all ranks considered while minimally affecting target performance.

|  | Target | Control |
|---|---|---|
| Pre-trained | 48.3 | 48.3 |
| Fine-tuned | 90.5 | 38.0 |
| Fine-tuned +LiNeS | 90.3 | 48.0 |
| $r = 16$ | 84.4 | 44.2 |
| $r = 16 + $ LiNeS | 84.3 | 46.7 |
| $r = 32$ | 85.8 | 42.8 |
| $r = 32 + $ LiNeS | 85.5 | 46.7 |
| $r = 64$ | 86.6 | 41.6 |
| $r = 64 + $ LiNeS | 86.4 | 46.2 |
| $r = 128$ | 87.5 | 41.6 |
| $r = 128 + $ LiNeS | 87.2 | 46.3 |

The single-task performance is presented in Figure 13 for ViT-B/32, Figure 14 for ViT-B/16, and Figure 15 for ViT-L/14. From the results we observe that our method demonstrates a noticeable improvement over baseline merging techniques across individual tasks in all test scenarios.

**Natural Language Processing** We provide in Figure 16 the detailed single-task performance for the three NLP benchmarks using T5-large, complementary to the results in Table 3. Similar to the observation in vision, our method provides a consistent improvement over baselines across individual tasks.

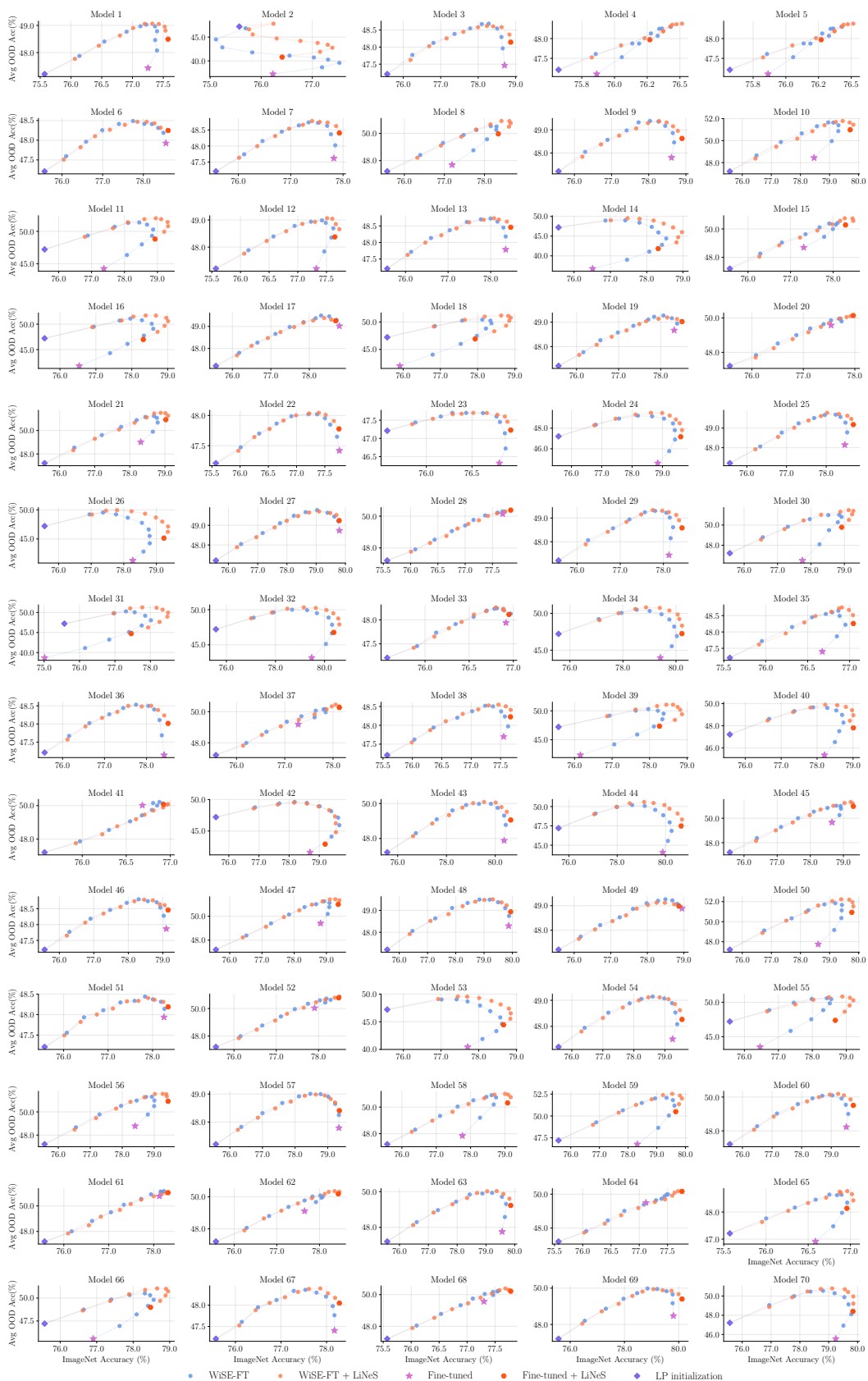

Figure 12: Performance of applying `LiNeS` to WiSE-FT to each ViT-B/32 checkpoint fine-tuned on ImageNet.

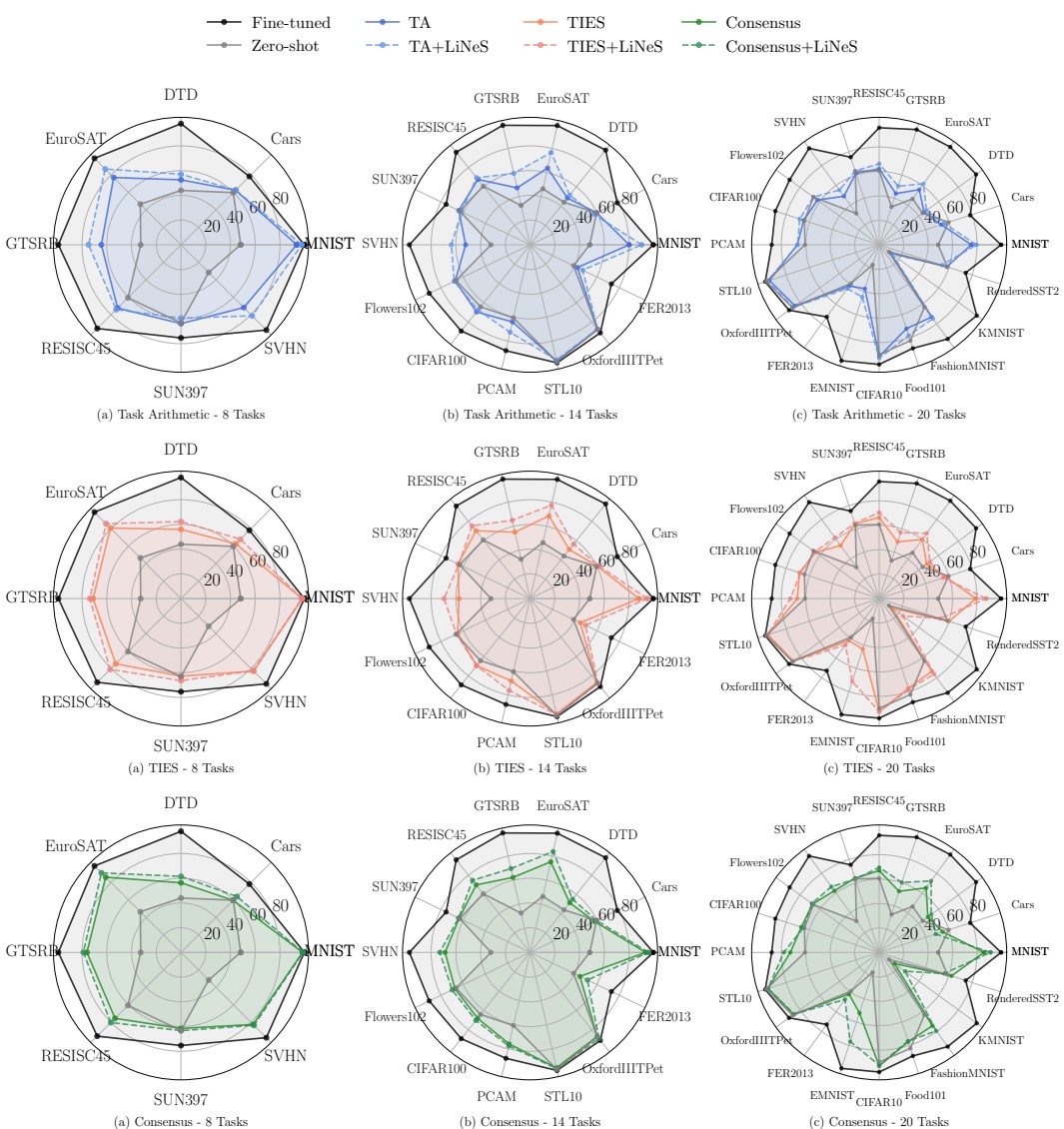

Figure 13: Single-task accuracies for multi-task merging on image classification benchmarks for ViT-B/32.

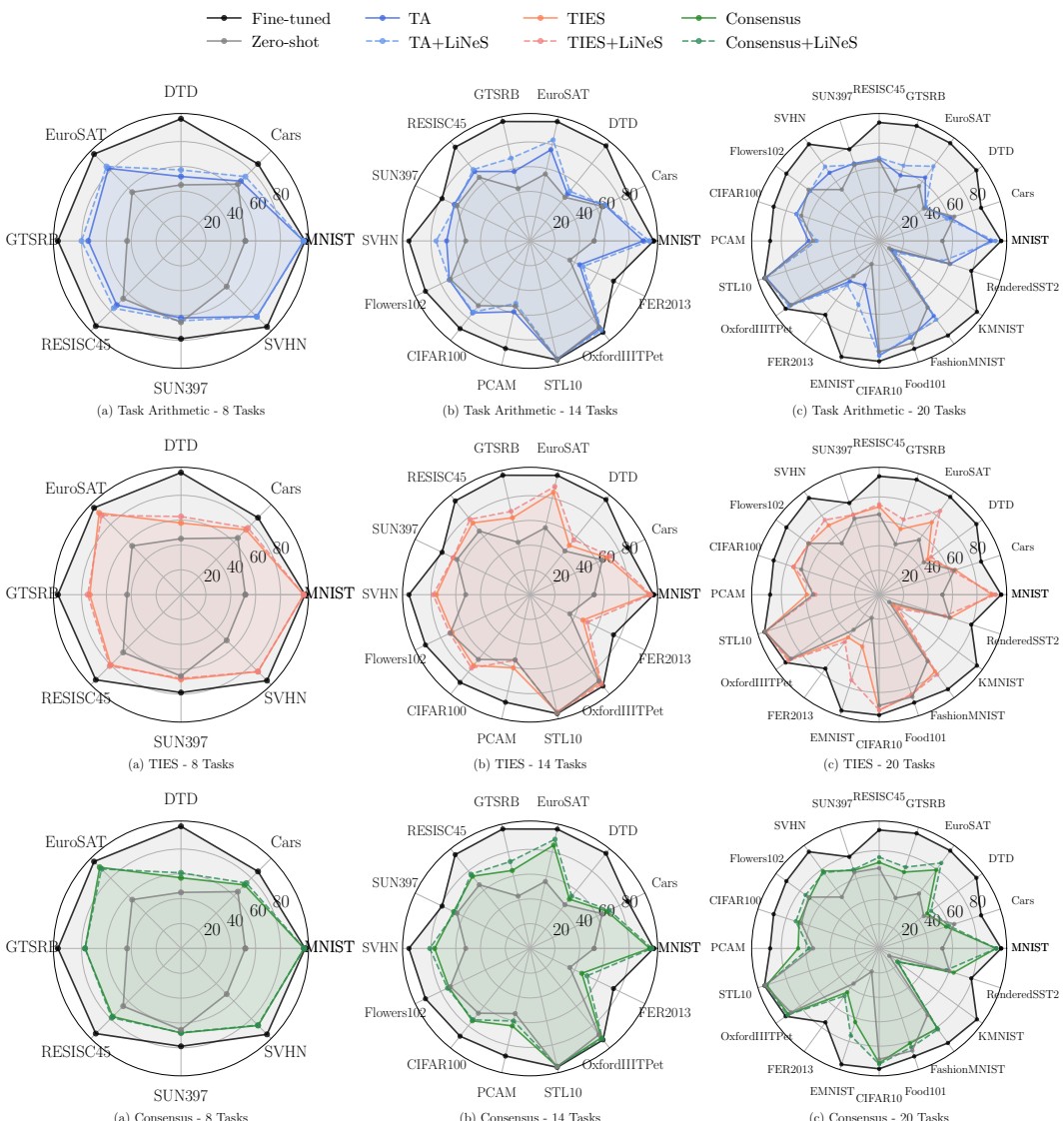

Figure 14: Single-task accuracies for multi-task merging on image classification benchmarks for ViT-B/16.

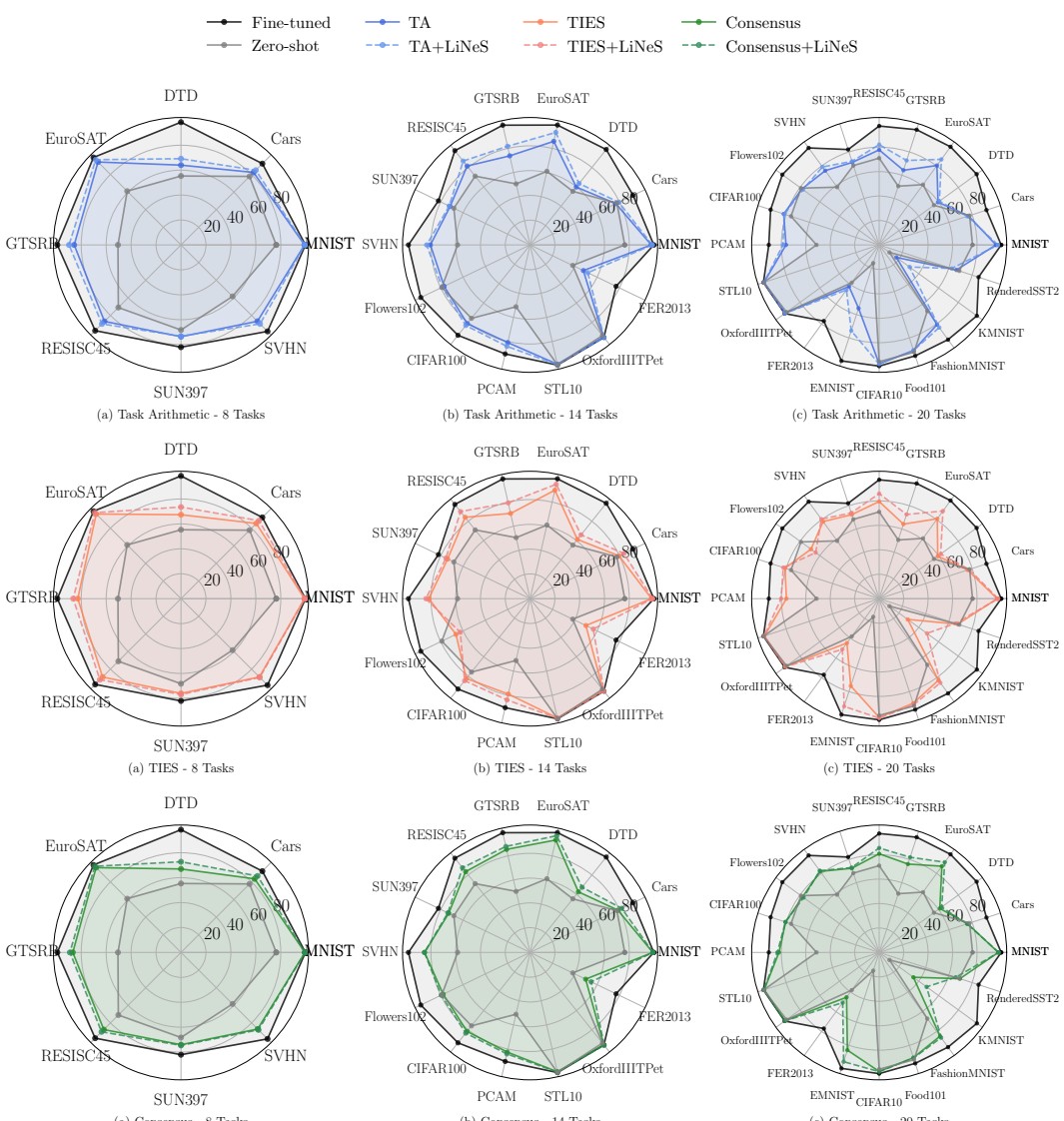

Figure 15: Single-task accuracies for multi-task merging on image classification benchmarks for ViT-L/14.

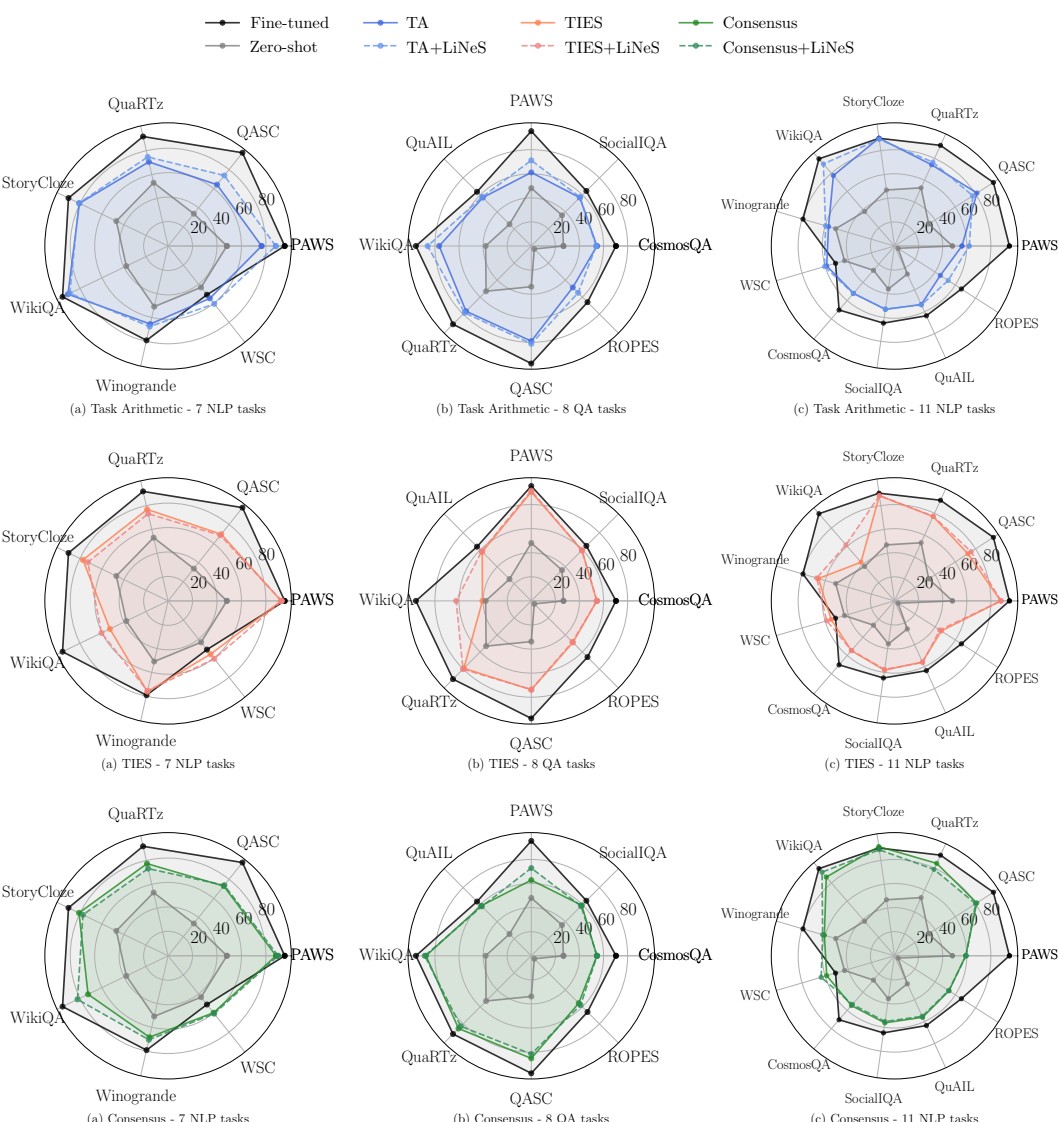

Figure 16: Single-task accuracies for multi-task merging on NLP benchmarks for T5-large.

