# OpenReview forum: "LiNeS: Post-training Layer Scaling Prevents Forgetting and Enhances Model Merging"
_ICLR.cc/2025/Conference — ICLR 2025 Poster_

### Official Review · Reviewer_tFK1 · 2024-11-01

**Soundness:** 3
**Presentation:** 3
**Contribution:** 2
**Rating:** 5
**Confidence:** 4

**Summary:**

The paper introduces a post-training technique designed to mitigate catastrophic forgetting; more specifically preserving pre-trained generalization while enhancing fine-tuned task performance. This is done by reducing the magnitude of parameter updates for the shallow-layers compared to the deeper-layers. The method boils down to a layer-wise linear interpolation between the fine-tuned and the pretrained model.

**Strengths:**

1) Overall the idea of the paper is pretty simple and should be easily reproducible. Further, the paper is clearly structured and mostly well-written.
2) The paper thoroughly evaluates the proposed method across multiple tasks.
3) The results clearly demonstrates strong trade-off performance on both target and control tasks.

**Weaknesses:**

1) Theoretical analysis have not been provided to support the proposed idea.
2) Some results lack details. For example, Table 1 what is the selected target and control tasks? Were all other combinations tried out? Not enough baselines have been added. Comparing to pretraining and fine-tuning performances is not enough.
3) How does learning the scaling factors perform compared to linearly scaling the parameter updates? Moreover, does the learned parameters have a consistent increasing trend from shallow-to-deep layers?

**Questions:**

1) Why a shallow layer must be updated less compared to a deeper layer? Theoretical analysis should be provided to why shallow layers need to be updated less than deeper layers and how does this method affects the model's representation space?
2) Clarify exactly which tasks were used as target and control in your results, and to explain their task selection criteria. Add more baselines that aims at protecting both generalization and the fine-tuning to a downstream task. This would clarify the performance improvements?
3) Conduct an ablation study comparing the linear scaling approach to a learned scaling approach, and analyze the resulting patterns in the learned scaling factors.
4) How does the proposed method differ from applying different learning rates per layer (low LR for shallow layers and higher LR for deep layers) during fine-tuning? Include a comparison between the proposed post-training method and the layer-wise learning rate scheduling during fine-tuning. This would help clarify the unique benefits of the proposed approach.
5) Provide an experiment to compare your proposed approach to freezing different shallow layers and updating the deeper layers using fine-tuning. This includes, but not limited to, freezing the feature extractor and finetuning the classifier.

---

> ### Author Response · Authors · 2024-11-19
>
> We thank reviewer tFK1 for their constructive feedback. We are pleased that they found our paper clearly structured and well-written, our evaluation thorough, and that our work demonstrates a strong performance tradeoff between target and control domains. We respond to their comments and questions below.
>
> **Theoretical analysis (W1 + Q1)**
>
> LiNeS is motivated by prior research and substantiated by strong and diverse empirical evidence. We acknowledge the reviewer's request for theoretical analyses but note that, given the scale of the models we consider (e.g., as large as LLaMa-7B), there are currently no formal mathematical tools or theoretical frameworks suitable for providing rigorous guarantees. In general, the field of model merging has yet no theoretical guarantees since the assumptions would likely be too restrictive to apply in the present practical settings.
>
>
> Therefore, a theoretical analysis is beyond the scope of this paper. However, we are open to considering any specific theoretical directions from the reviewer for future work. It is also important to recognize that our method is well-motivated, following a long line of work in understanding neural networks and the effect of each layer, see paper references: Neyshabur et al., 2020; Yosinski et al., 2014; Raghu et al., 2019b, as well as multi-task model merging works, such as Ilharco et al., 2023; Yadav et al., 2023, Wang et al., 2024, among others. Given that this line of work does not have any theory and given our strong empirical evidence across an extensive and diverse list of settings, we kindly encourage the reviewer to reconsider their score.
>
> **Clarification on Task Combination (W2 + Q2)**
>
> For Table 1, we used a round-robin approach, testing all possible combinations of target and control tasks. Whenever one of the 8 tasks is selected as the target task, the remaining 7 tasks are set as control tasks. The final results in Table 1 are averaged over the 8 task combinations, covering all possible combination scenarios. It is exactly our intention to demonstrate the robustness and general applicability of our method by testing it on all possible combinations. We have added a sentence to the paper to emphasize this point.
>
>
> **Learned Scaling (W3+Q3)**
>
> The original submission already includes a detailed comparison with methods that learn scaling coefficients, which eventually display consistently increasing behaviors, as discussed in Section 6. Specifically, we evaluated Ada-Merging, an unsupervised approach that leverages entropy minimization to learn these coefficients. On the 8-task benchmark, Ada-Merging reported 81.1% averaged multi-task accuracy. However, this comes with significant trade-offs in resource requirements and unreasonable assumptions. In particular, Ada-Merging assumes that all task vectors are available at test time, breaking the model merging rationale. Moreover, it loads all task vectors simultaneously, resulting in a linear increase in memory consumption with the number of tasks. In contrast, LiNeS leverages insights on the hierarchical nature of learned features, constraining shallow layers to preserve general features. This allows our method to scale efficiently to large models like LLaMA (Section 5.4) or a large number of tasks (Section 5.2). Furthermore, defining an unsupervised loss introduces additional complexity, limiting its broader applicability, especially in language settings. In fact, Ada-Merging has been only applied to computer vision classification tasks.
>
> Moreover, we also considered aTLAS, a contemporaneous work that learns scaling factors in a supervised manner using cross-entropy, which suffers from computational limitations similar to those of Ada-merging. Remarkably, as shown in Figure 6, the learned scalings for both methods demonstrate a clearly increasing trend, which is very close to those obtained without training by LiNeS.

---

> > ### Author Response · Authors · 2024-11-19
> >
> > **Fine-tuning based baselines (Q4 + Q5)**
> >
> > *Differences to fine-tuning based methods*
> >
> > LiNeS, as a post-training editing method, does not require modification to the fine-tuning process and is meant to address many of the limitations of standard fine-tuning by being highly resource-efficient and scalable to large-scale scenarios. Indeed, LiNeS eliminates the need for costly re-training and extensive hyperparameter tuning, operating directly on fine-tuned checkpoints without requiring access to the original training data or multiple iterative runs. Therefore, while mitigating forgetting can be achieved by altering fine-tuning, e.g., modifying the learning rate per layer, this process introduces numerous hyper-parameter choices on top of the other standard ones involved during fine-tuning, making it inefficient.
> >
> > *Performance Comparison*
> >
> > Following the reviewer’s suggestions, we present different modifications of the learning rate per layer.
> > We note that following previous works, we focus on the open-vocabulary setting. Specifically, for CLIP, the text encoder is frozen, acting as a classification layer, and we only fine-tune the visual encoder. For this reason, linear probing would result in modifying the text encoder or adding a randomly initialized linear layer.  Both settings would make the model close-vocabulary, and, for this reason, we do not consider linear probing.
> > Instead, we consider an alternative to tuning only the final block of the visual encoder, freezing all the other layers during training.
> >
> > Overall, we compare applying LiNeS with the following baselines to regularize the fine-tuning process:
> >
> > 1. **Fine-tuning with Linear Layer-Wise Learning Rate Decay (LinLR):**  Applies a linear learning rate schedule where the learning rate linearly increases from 0.0 to maximum value over all the layers.
> >
> > 2. **Fine-tuning with Exponential Layer-Wise Learning Rate Decay (ExpLR):**  Applies an exponential learning rate schedule where the learning rate is set to maximum for the deepest layers and decays by a factor of 0.5 by each layer for the shallower layers.
> >
> > 3. **Fine-tuning with First Half of Blocks Frozen (HalfFT):**  Freezes the parameters of the first half of the encoder blocks during training.
> >
> > 4. **Fine-tuning only the Final Block (LastFT):**  Freezes all blocks except the final one of the feature encoder.
> >
> > The result comparison is listed below:
> >
> >
> > | Method         | Target (%) | Control (%) |
> > |----------------|------------|-------------|
> > | Pre-trained    | 48.3       | 48.3        |
> > | Fine-tuned     | 90.5       | 38.0        |
> > | FT + LiNeS     | 90.3       | 48.0        |
> > | FT + LinLR     | 90.7       | 46.9        |
> > | FT + ExpLR     | 89.6       | 46.0        |
> > | FT + HalfFT    | 90.4       | 46.8        |
> > | FT + LastFT    | 86.6       | 46.6        |
> >
> >
> > From the results, we observe that LiNeS, as a post-training editing method, outperforms the regularized fine-tuning methods in terms of restoring the zero-shot performance on the control tasks. We have incorporated these additional baselines into the manuscript in Appendix C.8. Given this new extensive evaluation showing the superiority of LiNeS in efficiency, flexibility, computational cost, and mitigating forgetting. We kindly ask the reviewer to reconsider their score.

---

> > > ### Comment · Reviewer_tFK1 · 2024-11-28
> > >
> > > Thank you for detailed response and your additional experiments.
> > >
> > > FT + LinLR seems to have a challenging performance compared to linearly decreasing $\lambda$, and actually compared to FT has maintained the target accuracy and boosted the control accuracy. This may indicate that with proper fine-tuning of the learning rate you can reach same performance. I understand that this is a post-training approach, but the proposed approach is basic with no strong reasonings.
> > >
> > > I will maintain my score.

---

> > > > ### Author Response · Authors · 2024-11-29
> > > >
> > > > We thank the reviewer for their answer, but we respectfully disagree with their comment. The reviewer (i) is focusing only on a small part of the submission, ignoring **all** our results on model merging scenarios, which consist of a large part of our submission and represent an increasingly important area for the community. (ii) They penalize us for providing a simple and actionable algorithm that covers many practical cases.
> > > >
> > > > According to the reviewer, the main problem with our paper is that the proposed algorithm is “basic with no strong reasonings”. We do not think these are grounds for rejection;
> > > >
> > > > **Simplicity** is an advantage and not a weakness. In fact, our work builds on “simple algorithms” **on which we actually improve in all cases**:
> > > > 1. Robust fine-tuning [1], as presented in Sec. 5.1, is a CVPR oral and best paper finalist, while an extremely simple idea of linearly interpolating between the pre-trained and fine-tuned models.
> > > > 2. Task Arithmetic [2], as presented in Sec. 5.2,  simply ensembles the weights of the models. Our method achieves **uniformly better results** in both NLP and vision benchmarks while having the exact same computational demands.
> > > > 3. Model Soups [3], as presented in Sec. 5.3, simply averages 71 checkpoints and is an influential paper in the field with 800+ citations.
> > > > 4. Rewarded Soups [4], as presented in Sec. 5.4,  simply interpolates between models fine-tuned with different rewards.
> > > >
> > > > Most of this line of work has no theoretical guarantees.
> > > >
> > > > **Layer-wise FT.** To the best of our knowledge, we are the first to show that this fine-tuning paradigm mitigates catastrophic forgetting for SOTA vision-language models. In other words, we have provided a positive result that goes beyond the model editing field where cases 1-4 operate. We view this as an additional contribution rather than a weakness of the paper.
> > > >
> > > > As we clearly already answered in our previous response, It is also important to note that performing hyper-parameter search on the layer-wise learning rate schedule is:
> > > > 1. Beyond the scope of the paper, where we mainly focus on model editing settings where checkpoints are assumed to be given.
> > > > 2. Impossible in cases where FT checkpoints are given, or there is no data or compute to fine-tune from scratch, e.g., plenty of checkpoints from huggingface come with no private training data. LiNeS can be used in these cases
> > > > 3. Much more computationally expensive and thus prohibitive in cases such as LLaMA models with billions of parameters (Sec. 5.4) or for Model Soups with 70 fine-tuned checkpoints (Sec. 5.3).
> > > >
> > > > In fact, we directly used several publicly available checkpoints for our experiments with LiNeS: 60 ViT checkpoints of different sizes and fine-tuned on different tasks, 11 T5-Large checkpoints fine-tuned on different tasks, 71 ViT models fine-tuned on ImageNet with different hyperparameters (including learning rate). It is impractical and extremely computationally expensive to perform a hyper-parameter search for layer-wise learning rates in all these cases. This realization highlights the flexibility and importance of our proposed method in real-world scenarios.
> > > >
> > > >
> > > > [1] Wortsman et al. Robust fine-tuning of zero-shot models. CVPR 2022
> > > >
> > > > [2] Wortsman et al. Model soups: averaging weights of multiple fine-tuned models improves accuracy without increasing inference time. ICML 2022.
> > > >
> > > > [3] Ilharco et al. Editing Models with Task Arithmetic. ICLR 2023
> > > >
> > > > [4] Ramé et al. Rewarded soups: towards pareto-optimal alignment by interpolating weights fine-tuned on diverse rewards. NeuIPS 2023.

---

### Official Review · Reviewer_McJs · 2024-11-03

**Soundness:** 3
**Presentation:** 4
**Contribution:** 3
**Rating:** 6
**Confidence:** 3

**Summary:**

This paper presents a novel post-training editing technique called LiNeS, designed to address the problem of catastrophic forgetting in large pre-trained models during fine-tuning. Catastrophic forgetting refers to the phenomenon where a model loses its generalization ability on other tasks when fine-tuned for a specific task. LiNeS balances the trade-off between retaining the generalization capability of the pre-trained model and enhancing performance on specific tasks through hierarchical scaling of parameter updates.

**Strengths:**

- The LiNeS method introduces a novel post-training editing technique that prevents catastrophic forgetting through hierarchical scaling of parameter updates.
- Extensive experiments validate the effectiveness of LiNeS across various scenarios, including single-task fine-tuning, multi-task model merging, and improved out-of-distribution (OOD) generalization.
- The LiNeS method is applicable not only to visual tasks but also to natural language processing tasks, demonstrating its cross-domain versatility.
- LiNeS can be integrated with existing multi-task model merging baselines.

**Weaknesses:**

- The article does not discuss the performance of LiNeS in long-term maintenance and updating of models, particularly when faced with evolving data distributions.
- There is a lack of experimental results for LLM tasks.
- Comparisons with the performance of other fine-tuning methods (such as LoRA) are missing.

**Questions:**

- Are there additional experimental results for LLMs, including different models, datasets, and tasks?

---

> ### Author Response · Authors · 2024-11-19
>
> We thank reviewer McJs for their constructive feedback. We are pleased that they found our proposed method novel, effective through extensive experiments, versatile across domains, and compatible with existing model merging baselines. We respond to their comments and questions.
>
> **Continual Training**
>
> We thank the reviewer for suggesting this potential area for exploration. In this paper, we focused on a static setting and demonstrated the effectiveness of LiNeS in mitigating forgetting of general knowledge after fine-tuning on specific tasks. Besides, as appreciated by the reviewer, we have conducted extensive experiments in demonstrating the effectiveness in many diverse problems in the literature, including OOD generalization, multi-task model merging, improving model soups, and merging LLMs fine-tuned with different rewards. Extending LiNeS to address long-term model maintenance and adaptation to evolving data distributions is beyond the scope of this paper while indeed an important direction for future work.
>
> **LLM experiments**
>
>
> We respectfully disagree on the lack of NLP results. In particular, we already tested against all the benchmarks explored by the field [1-3], containing tasks such as QASC, QuaRTz, PAWS, Story Cloze, WikiQA, Winogrande, WSC, CosmosQA, QuAIL, ROPES, and SocialIQA using the T5-large language model which is widely used in this literature [1-4]. Furthermore, beyond these previously explored tasks, we are among the first works in the model merging field to experiment on LLaMA-2 (with 7B parameters), following the setup of Rewarded Soups [5], and for which we fine-tune the checkpoints using PPO. Specifically, we apply LiNeS to improve the merging of two LLMs fine-tuned with 2 different rewards, leading to a merged model with a dominating Pareto-Front on both rewards compared to the model obtained with rewarded soups. The experiments in the paper are extensive, covering different applications and different modalities. We hope this clarifies the reviewer’s comment, and therefore, we encourage them to reassess their score.
>
> **LoRA and Fine-tuning baselines**
>
> We thank the reviewer for their suggestion. We note that LiNeS operates as a post-training editing method, making it orthogonal and compatible with standard fine-tuning methods such as LoRA. We have shown LiNeS’s compatibility and orthogonality with LoRA in the experiment for merging LLaMA-7B models (Section 5.4), where they are fine-tuned exactly with LoRA. We made this point more clear in the revised version of the paper.
>
> Moreover, following the reviewer’s suggestion, we have added additional results for models fine-tuned with LoRA in the updated manuscript (Appendix C.8.4), with a focus on the mitigation of catastrophic forgetting. We have considered different hyper-parameter settings for LoRA. The results in Table 11 show that the models fine-tuned with LoRA also demonstrate degraded performance on control tasks to as much as 6.4\%. Nevertheless, we further show that LiNeS can be applied to models after fine-tuning with LoRA to mitigate catastrophic forgetting. For example, for the models fine-tuned with LoRA with rank=128, LiNeS effectively improves the control task performance from 41.6\% to 46.3\% while preserving almost full performance in target tasks (87.2\% out of 87.5\%). The additional results demonstrate LiNeS’s orthogonality and compatibility with parameter-efficient fine-tuning methods like LoRA.
>
> For convenience, we add the table regarding experiments with LoRA below, where $r$ indicates the different ranks:
>
> |                       | Target | Control |
> |-----------------------|--------|---------|
> | **Pre-trained**       | 48.3   | 48.3    |
> | **Fine-tuned**        | 90.5   | 38.0    |
> | **Fine-tuned + $LiNeS$** | 90.3   | 48.0    |
> | **$r=16$**            | 84.4   | 44.2    |
> | **$r=16 + LiNeS$** | 84.3   | 46.7    |
> | **$r=32$**            | 85.8   | 42.8    |
> | **$r=32 + LiNeS$** | 85.5   | 46.7    |
> | **$r=64$**            | 86.6   | 41.6    |
> | **$r=64 + LiNeS$** | 86.4   | 46.2    |
> | **$r=128$**           | 87.5   | 41.6    |
> | **$r=128 + LiNeS$** | 87.2   | 46.3    |
>
> Finally, as suggested by other reviewers, we have also added experiments with different fine-tuning schemes beyond standard and LoRA. Specifically, we considered layer-wise learning rate decays, linear and exponential, as well as freezing early layers of the network. The results are also presented in Appendix C.8 of the updated manuscript. It is important to note that modifying the fine-tuning process introduces multiple hyperparameters, and each experiment is computationally expensive and time-consuming. In contrast, LiNeS is flexible and efficient while achieving the least forgetting, as shown by the added results.

---

> > ### Author Response · Authors · 2024-11-19
> >
> > [1] Yadav et al. Ties-merging: Resolving interference when merging models. NeurIPS 2023
> >
> > [2] Wang et al.  Localizing Task Information for Improved Model Merging and Compression. ICML 2024
> >
> > [3] Tam et al. Merging by Matching Models in Task Parameter Subspaces. TMLR 2024
> >
> > [4] Ilharco et al. Editing Models with Task Arithmetic. ICLR 2023
> >
> > [5] Ramé et al. Rewarded soups: towards pareto-optimal alignment by interpolating weights fine-tuned on diverse rewards. NeuIPS 2023.

---

> ### Comment · Reviewer_McJs · 2024-11-26
>
> Thank you for your response. I have decided to maintain my rating score.

---

> > ### Author Response · Authors · 2024-11-26
> >
> > Dear reviewer McJs,
> >
> > Thank you for your time reviewing our paper.
> >
> > In our rebuttal, we have addressed the concerns raised in the original review. It is important to note that extensive LLM experiments (W2) were already present in the manuscript. Furthermore, we strongly believe that we have addressed W3 (“fine-tuning baselines”) by including several experiments with LoRA fine-tuning during the rebuttal, where we show that our method still delivers strong performance gains as for SFT. Overall, our method, when combined with full fine-tuning, dominates LoRA-based approaches by outperforming them on both target and control task accuracies.
> >
> > Could you please point us to any remaining limitations of our work? We would be happy to address any additional concerns in the extended discussion period.
> >
> > Best regards,
> >
> > the authors

---

### Official Review · Reviewer_KCXu · 2024-11-04

**Soundness:** 3
**Presentation:** 3
**Contribution:** 3
**Rating:** 6
**Confidence:** 4

**Summary:**

This paper introduces LiNeS, a post-training editing technique designed to enhance the generalization ability of fine-tuned models. LiNeS applies a layer-wise linear scaling function to modulate parameter updates, reducing them more in shallow layers than in deep layers. The method is evaluated across several settings to demonstrate its effectiveness.

**Strengths:**

- The paper is well-motivated and easy to follow.
- The experimental results seem impressive.

**Weaknesses:**

- The validation set has a large influence on the OOD performance of the selected models [1]. I suggest the authors discuss the construction of the validation set and whether the same empirical trend can be observed with different validation sets, as suggested by [1].
- The evaluation is limited to the transformer architecture. Including other model architectures, such as ResNet-based CLIP models, would validate whether the benefits of layer-wise scaling extend beyond transformer architecture.
- There is a lack of sensitivity analysis for the hyper-parameter $\alpha$ and $\beta$ across different evaluation settings.
- Typos should be checked. For example,
    - enchancing → enhancing (Line 23)
    - bcenchmarks → benchmarks (Line 24)
    - task=specific → task-specific (Line 112)
    - Figure 4 → Table 4

[1] In search of lost domain generalization. In ICLR, 2021

**Questions:**

- What could be an explanation for why the benefit of LiNeS appear to be smaller for deeper models (e.g. ViT-L/14 in Table 2) and models with smaller patch sizes (e.g. ViT-B/16 in Table 5)?

---

> ### Author Response · Authors · 2024-11-19
>
> We thank reviewer KCXu for their constructive feedback. We are pleased that they find our paper well-motivated and easy to follow, and our results impressive. We respond to each weakness and question below. The additional experiments requested by the reviewer can be found in the updated manuscript in Appendix C.8.
>
> **OOD Experiments**
>
> The evaluation of OOD performance (see Section 5.1) is performed on test sets without searching for hyper-parameters. Thus, there is no influence from a validation set. It is important to note that the benchmark and evaluation pipeline are not our own and are widely used in the literature and, more importantly, our baseline [1] [CVPR 2022 oral](https://cvpr2022.thecvf.com/orals-622-pm). Additionally, our paper does not solely focus on OOD generalization but showcases an issue found in many settings that our method resolves. Beyond OOD generalization, we also include experiments on multi-task model merging (vision and NLP), single-task model merging, and merging foundation models fine-tuned on different rewards.
>
> **Beyond Transformer Architectures**
>
> In all our experiments, we follow the experimental protocols established by prior works that only use transformer architectures, which currently dominate both the CV and NLP landscape. Specifically, §5.1 comes from [1], §5.2 from [3-5], §5.3 from [2], and §5.4 from [6] and [7]. The inclusion of so many diverse settings where our method improves performance highlights the generality of LiNeS.
>
> Given the reviewer's suggestion and to further validate the robustness of our findings, we have explored different architectures, such as CNNs. ResNet-based architectures have been shown to not work well on task arithmetic scenarios (see, e.g., [9]), probably due to a lack of sufficient weight disentanglement [8]. Yet, we have added new results on ConvNext, following [8], in Appendix C.8.2. Remarkably, LiNeS improves performance in both mitigating catastrophic forgetting (similar to §3) as well as multi-task model merging scenarios (similar to §5.2). Given these new strong results addressing the reviewer's concern, we kindly ask them to reconsider their score.
>
> **Sensitivity analysis**
>
> We thank the reviewer for the suggestion. We have added new experiments focusing on the sensitivity w.r.t. the hyperparameters in Appendix C.8.1.
>
> We emphasize that for multi-task merging, $\alpha$ is not tuned as a hyper-parameter but fixed via a heuristic to keep the computational cost the same as the baseline methods, as explained in §5.2. Therefore, we focus on the hyperparameter $\beta$ – the slope of the line – and compare its sensitivity to the uniform scaling hyperparameter $\lambda$ used by prior work such as Task Arithmetic, Ties, and Consensus. Our results, in Figure 14, show that LiNeS is less sensitive compared to the corresponding constant scaling coefficient of vanilla Task Arithmetic while improving performance.
>
> Furthermore, we also conducted an experiment to test the sensitivity of the multi-task performance to both $\alpha$ and $\beta$, treating them both as hyper-parameters and presenting the results as a 2D heatmap in Figure 15, Appendix C.8.1. The heatmap clearly demonstrates the necessity for applying a layer-increasing scaling schedule for multi-task merging, as the optimal performances are obtained with $\beta>0$ for all three merging methods. In these new results, we also indicate the optimal configuration, which validates the heuristics previously introduced in the paper (cf. Eq. 2) to restrict hyperparameter search to a single parameter, i.e., the slope. Notably, with the latter choice, LiNeS maintains the computational complexity of the baseline methods.
>
> **Typos**
>
> We have updated the manuscript to correct the typos, we thank the reviewer for highlighting them.
>
> **Q1**
>
> This phenomenon is observed across the multi-task model merging literature [3,4,5,7]. The reason is that larger models are more weight-disentangled [8], i.e., the task representations in weight space are more disentangled and models can retain a larger portion of the performance before merging. For instance, vanilla task arithmetic [3] drops from 90.5% fine-tuning accuracy to 69.7% for ViT-B-32 and 8 tasks (see Table 2) while for 8 tasks with ViT-L-14 the drop is from 94% to 84%, and, therefore, any method including ours has less room for improvement over [3] with the larger models.
>
> Importantly, we note that LiNeS outperforms the previous methods with larger models. For example, for ViT-L-14 architecture, Ties-merging improves Task Arithmetic respectively by 1.6%, 0.1%, 1.6% for the 3 vision benchmarks, while LiNeS improves Task Arithmetic by 2.5%, 3.0%, 3.1% respectively, and it further improves Ties-merging by 2.4%, 3.2%, 4.0% respectively.

---

> > ### Author Response · Authors · 2024-11-19
> >
> > [1] Wortsman et al. Robust fine-tuning of zero-shot models. CVPR 2022
> >
> > [2] Wortsman et al. Model soups: averaging weights of multiple fine-tuned models improves accuracy without increasing inference time. ICML 2022.
> >
> > [3] Ilharco et al. Editing Models with Task Arithmetic. ICLR 2023
> >
> > [4] Yadav et al. Ties-merging: Resolving interference when merging models. NeurIPS 2023
> >
> > [5] Wang et al. Localizing Task Information for Improved Model Merging and Compression. ICML 2024
> >
> > [6] Ramé et al. Rewarded soups: towards pareto-optimal alignment by interpolating weights fine-tuned on diverse rewards. NeuIPS 2023.
> >
> > [7] Yang et al. Rewards-in-Context: Multi-objective Alignment of Foundation Models with Dynamic Preference Adjustment. ICML 2024.
> >
> > [8] Ortiz-Jimenez et al. Task arithmetic in the tangent space: Improved editing of pre-trained models. NeurIPS 2023
> >
> > [9] Ilharco et al. Patching open-vocabulary models by interpolating weights. NeurIPS 2022.

---

> > > ### Comment · Reviewer_KCXu · 2024-12-01
> > >
> > > Most of my concerns have been addressed, and I have decided to increase my score to 6.

---

> > > > ### Author Response · Authors · 2024-12-01
> > > >
> > > > We are glad that the reviewer’s concerns have been addressed and we thank them for increasing their score.
> > > >
> > > > Kind regards,
> > > >
> > > > the authors

---

> ### Comment · Area_Chair_QbVo · 2024-11-30
>
> Dear Reviewer,
>
> Could you kindly respond and indicate whether authors have addressed your concerns?
>
> Thanks, AC

---

### Official Review · Reviewer_oncK · 2024-11-09

**Soundness:** 3
**Presentation:** 2
**Contribution:** 3
**Rating:** 6
**Confidence:** 3

**Summary:**

This paper introduces a post-training editing technique, LiNeS, designed to address catastrophic forgetting and facilitate model merging after fine-tuning. LiNeS scales parameter updates linearly with the depth of layers within the network. The technique has shown significant improvements across various domains, including OOD generalization, single-task and multi-task model merging, demonstrating its effectiveness.

**Strengths:**

1. The paper is easy-to-follow.

2. The topic is essential yet the idea is moderate. Both regularized fine-tuning and model merging are important techniques for the community and the paper addresses two formulations simultaneously.

3. A new method that seems well-motivated and performs well on a lot of benchmarks.

**Weaknesses:**

1. I like the fact that the authors simply scale the task vectors. But I was not clear why directly edits the difference between the fine-tuned and pre-trained checkpoint via a linearly scaling coefficient ($\alpha+\beta\frac{l-1}{L-1}$). Also, compare to existing methods, like `Ties-Merging` and `Consensus Merging`, what are the core strengths of this work? More efficient or simpler?

2. The point above also points to the limitation of the current framework, which is a lack of formal theory. It works very well experimentally, but some aspects are not very clear. Providing concrete theoretical analyses or proofs for linearly layer-wise scaling would be better.

3. There are numerous typos and bad grammar throughout the paper. The authors should do a very careful proofreading and fix all the errors.

4. Minor comments:
- Some of the content in the appendix can be put into the body to ensure that the body is a full 10 pages.
- It must be noted that the consistency in verb tense is not maintained throughout the document. For example, the second paragraph in the Introduction section.
- line 112 "task=specific"
-  Two identical sentences appear from line 223 to line 226.

**Questions:**

1. The basic assumption of this work is that the degradation of performance on control tasks is largely due to these distortions in the shallow layers. However, it lacks of formal theory.

2. From Fig. 2 and Fig. 7, fine-tuned models have obtained high performance for both target task and control tasks in some cases. Can catastrophic forgettinh happen when the result is high?  Please discuss or analyze these specific cases where fine-tuning seems to improve performance on both target and control tasks.

3. The result of "Model 2" in Fig. 10 is different from the other 69 curves, what is the reason?

---

> ### Author Response · Authors · 2024-11-19
>
> We thank reviewer oncK for their valuable feedback. We are pleased they find our method well-motivated,  well-benchmarked, and the paper easy to follow. Below, we address their specific comments and questions.
>
>
> **W1**
>
>
> LiNeS is a post-processing method that can be applied to any single- or multi-task vector, making it orthogonal to existing and future model merging methods, such as `Ties` and `Consensus Merging`. This is a major strength of our approach. As demonstrated in Tables 2, 3, and 5, our method delivers performance increases in all 36 benchmarks, covering the combinations of underlying methods, baselines, and models. These improvements are achieved on top of the baseline methods, including `Ties` and `Consensus Merging`.  Furthermore, LiNeS is simpler than `Ties` and `Consensus` and often yields better results than them individually when applied to vanilla task arithmetic.
>
> Our choice of increasing schedule is motivated by prior literature, which indicates that general features reside in the first layers, whereas task-specific features are predominantly found in the deep layers. This insight is corroborated by our empirical findings in Section 3, where we show that linearly scaling updates effectively mitigates catastrophic forgetting. We opted for linear scaling since it is the simplest design choice. While we also explored alternative scaling functions, such as square root and quadratic (Table 4 in the Appendix), we observed comparable performance. Investigating more sophisticated scaling choices – which would likely require more hyperparameters than LiNeS – is an interesting direction that we leave for future research.
>
>
> **W2**
>
>
> LiNeS is motivated by prior research and substantiated by strong and diverse empirical evidence. We acknowledge the reviewer's request for theoretical analyses but note that, given the scale of the models we consider (e.g., as large as LLaMa-7B), there are currently no formal mathematical tools or theoretical frameworks suitable for providing rigorous guarantees. In general, the field of model merging has yet no theoretical guarantees since the assumptions would likely be too restrictive to apply in the present practical settings.
>
>
> Therefore, a theoretical analysis is beyond the scope of this paper. However, we are open to considering any specific theoretical directions from the reviewer for future work. It is also important to recognize that our method is well-motivated, following a long line of work in understanding neural networks and the effect of each layer, see paper references: Neyshabur et al., 2020; Yosinski et al., 2014; Raghu et al., 2019b, as well as multi-task model merging works, such as Ilharco et al., 2023; Yadav et al., 2023, Wang et al., 2024, among others. Given that this line of work does not have any theory and given our strong empirical evidence across an extensive and diverse list of settings, we kindly encourage the reviewer to reconsider their score.
>
>
> **Typos**
>
>
> Thank you for highlighting these issues – we have updated the manuscript to correct them.
>
>
> **Q1**
>
>
> The origin of the degradation of performance on control tasks is not just an assumption since (1) it is shown comprehensively in our motivating section 3 (Figures 1+2) and (2) is inspired by a long line of work showing similar observations, such as the paper references (Neyshabur et al., 2020; Yosinski et al., 2014; Raghu et al., 2019b).
>
>
> **Q2**
>
>
> On control tasks, we measure normalized accuracy with respect to the zero-shot performance of the model. All fine-tuned models (the blue marker points) in Figures 2 and 7 achieve less than 100% normalized control accuracy. Thus, there are no such cases as the ones mentioned by the reviewer where fine-tuning improves performance on control tasks. In fact, fine-tuning systematically decreases performance on the control tasks, referred to as `catastrophic forgetting`. It is only when applying our method on the fine-tuned checkpoints (in orange) that the performance is close to (or even surpasses) 100% on both target and control tasks. We also point out that, as suggested by reviewer KCXu, we have added new experiments showing the same results with convolutional architectures.
>
>
> **Q3**
>
>
> We note that all 70 models come directly from the published checkpoints of a previous work, Model Soups, fine-tuned with different hyperparameter configurations, available [in their repo](https://github.com/mlfoundations/model-soups/blob/main/hparam_info.json). Model 2 was trained with the largest learning rate and longest epochs, resulting in significant differences between the pre-trained and fine-tuned weights. As the weight interpolation literature suggests, these choices can break mode connectivity. Therefore, this issue originates from the specific hyperparameter choice and not our method.

---

> > ### Comment · Reviewer_oncK · 2024-12-01
> >
> > Thank you for your rebuttal. All of my concerns have been addressed. I'm slightly in favor of acceptance.

---

> > > ### Author Response · Authors · 2024-12-01
> > >
> > > We are glad that all the concerns of the reviewer have been addressed, and we thank the reviewer for supporting the acceptance of our work.
> > >
> > > Kind regards,
> > >
> > > the authors

---

> ### Comment · Area_Chair_QbVo · 2024-11-30
>
> Dear Reviewer,
>
> Could you kindly respond and indicate whether authors have addressed your concerns?
>
> Thanks, AC

---

### Author Response · Authors · 2024-11-19

We thank the reviewers for their constructive feedback. We are pleased that reviewers find our paper easy to follow (oncK, KCXu), well-written and clearly structured (tFK1), our method well-motivated (oncK, KCXu), novel (McJs), effective (oncK, KCXu), our experimental results and improvements significant (oncK), impressive (KCXu), extensive (McJs), thorough and strong (oncK).

We have included a response to each reviewer. Here, we briefly mention new experimental results following the reviewers’ suggestions. Changes in the manuscript are highlighted in blue font. All the following material was included in Appendix C.8 of the updated manuscript:

- **LoRA**: We added additional results for models fine-tuned with LoRA. We show that (1) models still suffer from catastrophic forgetting when fine-tuned with LoRA, and (2) LiNeS can be applied on top of LoRA, effectively restoring zero-shot abilities.

- **Regularized fine-tuning**: We included various baselines for layer-wise modifications to the fine-tuning process. LiNeS outperforms all these baselines in terms of restoring zero-shot performance. It is important to note that modifying fine-tuning is significantly more complex and computationally expensive than post-training methods such as LiNeS.

- **Convolutional architectures**: We included experiments with ConvNeXt architectures, demonstrating a similar forgetting issue after fine-tuning. LiNeS mitigates catastrophic forgetting and consistently improves multi-task model merging, showing similar results as the case with ViTs.

- **Sensitivity analysis to hyper-parameters**: We added a sensitivity analysis for the hyper-parameters of LiNeS in the multi-task merging setting. LiNeS is less sensitive to the hyper-parameters than the baseline model merging methods. We also reported the multi-task performance over all combinations of $\alpha$ and $\beta$, motivating the heuristics used in the paper and applying an increasing scaling schedule.

---

### Comment · Area_Chair_QbVo · 2024-11-28
**Dear reviewers, please kindly respond**

Dear Reviewers,

If you have not responded to author's rebuttal, please kindly do so as soon as possible. The deadline is Dec 2, but the authors can potentially further clarify questions if you respond earlier. Thanks!

Best, AC

---

### Meta-Review · Area_Chair_QbVo · 2024-12-23

**Metareview:**

Summary: LiNeS is a post-training method that applies layer-wise scaling of weight distances, preserving general features while enhancing task-specific deeper features in single- or multi-task settings. It can mitigate catastrophic forgetting and improve model merging.

Strengths: simple yet novel post-training technique; extensive experiments across vision and NLP; compatible with existing model merging baselines.

Weaknesses: only on transformer architectures; more baselines needed; lacking theoretical analysis (not required)

Reasons for decision: LiNeS' ability to effectively preserve pre-trained generalization while enhancing task performance through a simple, versatile post-training method outweighs its minor weaknesses above.

**Additional Comments On Reviewer Discussion:**

The authors addressed reviewer concerns, adding experiments for other architectures (e.g., ConvNeXt), sensitivity analysis for hyperparameters, and comparisons with fine-tuning methods (e.g., LoRA). They clarified task selection and baselines and improved presentation issues, leading to one reviewer upgrading the rating.

---

### Decision · Program_Chairs · 2025-01-22

Accept (Poster)